# Adaptive Learning of Rank-One Models for Efficient Pairwise Sequence Alignment

**Govinda M. Kamath**[*1], **Tavor Z. Baharav** [*2], **and Ilan Shomorony** [3]

[1]Microsoft Research New England, Cambridge, MA
[2]Department of Electrical Engineering, Stanford University, Stanford, CA
[3]Department of Electrical and Computer Engineering,
University of Illinois, Urbana-Champaign, IL

gokamath@microsoft.com, tavorb@stanford.edu, ilans@illinois.edu

## Abstract

Pairwise alignment of DNA sequencing data is a ubiquitous task in bioinformatics and typically represents a heavy computational burden. State-of-the-art approaches to speed up this task use hashing to identify short segments ($k$-mers) that are shared by pairs of reads, which can then be used to estimate alignment scores. However, when the number of reads is large, accurately estimating alignment scores for all pairs is still very costly. Moreover, in practice, one is only interested in identifying pairs of reads with large alignment scores. In this work, we propose a new approach to pairwise alignment estimation based on two key new ingredients. The first ingredient is to cast the problem of pairwise alignment estimation under a general framework of rank-one crowdsourcing models, where the workers' responses correspond to $k$-mer hash collisions. These models can be accurately solved via a spectral decomposition of the response matrix. The second ingredient is to utilise a multi-armed bandit algorithm to adaptively refine this spectral estimator only for read pairs that are likely to have large alignments. The resulting algorithm iteratively performs a spectral decomposition of the response matrix for adaptively chosen subsets of the read pairs.

## 1   Introduction

A key step in many bioinformatics analysis pipelines is the identification of regions of similarity between pairs of DNA sequencing reads. This task, known as *pairwise sequence alignment*, is a heavy computational burden, particularly in the context of third-generation long-read sequencing technologies, which produce noisy reads [45]. This challenge is commonly addressed via a two-step approach: first, an alignment estimation procedure is used to identify those pairs that are likely to have a large alignment. Then, computationally intensive alignment algorithms are applied only to the selected pairs. This two-step approach can greatly speed up the alignment task because, in practice, one only cares about the alignment between reads with a large sequence identity or overlap.

Several works have developed ways to efficiently estimate pairwise alignments [6, 29, 30, 36]. The proposed algorithms typically rely on hashing to efficiently find pairs of reads that share many $k$-mers (length-$k$ contiguous substrings). Particularly relevant to our discussion is the MHAP algorithm of Berlin et al. [6]. Suppose we want to estimate the overlap size between two strings $S_0$ and $S_1$ and let $\Gamma(S_i)$ be the set of all $k$-mers in $S_i$, $i = 0, 1$. For a hash function $h$, we can compute a min-hash

$$h(S_i) \triangleq \min\{h(x) : x \in \Gamma(S_i)\}, \tag{1}$$

---

[*]Equal Contributors

for each read $S_i$. The key observation behind MHAP is that, for a randomly selected hash function $h$,

$$\mathbb{P}\left[h(S_0) = h(S_1)\right] = \frac{|\Gamma(S_0) \cap \Gamma(S_1)|}{|\Gamma(S_0) \cup \Gamma(S_1)|}. \tag{2}$$

In other words, the indicator function $\mathbb{1}\{h(S_0) = h(S_1)\}$ provides an unbiased estimator for the $k$-mer *Jaccard similarity* between the sets $\Gamma(S_0)$ and $\Gamma(S_1)$, which we denote by $\mathrm{JS}_k(S_0, S_1)$. By computing $\mathbb{1}\{h(S_0) = h(S_1)\}$ for several different random hash functions, one can thus obtain an arbitrarily accurate estimate of $\mathrm{JS}_k(S_0, S_1)$. As discussed in [6], $\mathrm{JS}_k(S_0, S_1)$ serves as an estimate for the overlap size and can be used to filter pairs of reads that are likely to have a significant overlap.

Now suppose we fix a reference read $S_0$ and wish to estimate the size of the overlap between $S_0$ and $S_i$, for $i = 1, \ldots, n$. Assume that all reads are of length $L$ and let $p_i \in [0, 1]$ be the overlap fraction between $S_i$ and $S_0$ (i.e., the maximum $p$ such that a $pL$-prefix of $S_i$ matches a $pL$-suffix of $S_0$ or vice-versa). By taking $m$ random hash functions $h_1, \ldots, h_m$, we can compute min-hashes $h_j(S_i)$ for $i = 0, 1, \ldots, n$ and $j = 1, \ldots, m$. The MHAP approach corresponds to estimating each $p_i$ as

$$\hat{p}_i = \frac{1}{m} \sum\nolimits_{j=1}^{m} \mathbb{1}\{h_j(S_0) = h_j(S_i)\}. \tag{3}$$

In the context of crowdsourcing and vote aggregation [16, 19, 40], one can think of each hash function $h_j$ as a worker/expert/participant, who is providing binary responses $Y_{i,j} = \mathbb{1}\{h_j(S_0) = h_j(S_i)\}$ to the questions "do $S_i$ and $S_0$ have a large alignment score?" for $i = 1, \ldots, n$. Based on the binary matrix of observations $Y = [Y_{i,j}]$, we want to estimate the true overlap fractions $p_1, \ldots, p_n$.

The idea of jointly estimating $p_1, \ldots, p_n$ from the whole matrix $Y$ was recently proposed by Baharav et al. [5]. The authors noticed that in practical datasets the distribution of $k$-mers can be heavily skewed. This causes some hash functions $h_j$ to be "better than others" at estimating alignment scores. Hence, much like in crowdsourcing models, each worker has a different level of expertise, which determines the quality of their answer to all questions. Motivated by this, Baharav et al. [5] proposed a model where each hash function $h_j$ has an associated unreliability parameter $q_j \in [0, 1]$ and, for $i = 1, \ldots, n$ and $j = 1, \ldots, m$, the binary observations are modeled as

$$Y_{i,j} \sim \mathrm{Ber}(p_i) \vee \mathrm{Ber}(q_j), \tag{4}$$

where $\mathrm{Ber}(p)$ is a Bernoulli distribution with parameter $p$ and $\vee$ is the OR operator. If a given $h_j$ assigns low values to common $k$-mers, spurious min-hash collisions are more likely to occur, leading to the observation $Y_{i,j} = 1$ when $S_i$ and $S_0$ do not have an overlap (thus being a "bad" hash function). Similarly, some workers in crowdsourcing applications provide less valuable feedback, but we cannot know a priori how reliable each worker is.

A key observation about the model in (4) is that, in expectation, the observation matrix $Y$ is rank-one after accounting for an offset. More precisely, since $\mathbb{E}Y_{i,j} = p_i + q_j - p_i q_j = (1 - p_i)(q_j - 1) + 1$,

$$\mathbb{E}Y - \mathbf{1}\mathbf{1}^T = (\mathbf{1} - \mathbf{p})(\mathbf{q} - \mathbf{1})^T, \tag{5}$$

where $\mathbf{p} = [p_1, \ldots, p_m]^T$ and $\mathbf{q} = [q_1, \ldots, q_n]^T$. Baharav et al. [5] proposed to estimate $\mathbf{p}$ by computing a singular value decomposition (SVD) of $Y - \mathbf{1}\mathbf{1}^T$, and setting $\hat{\mathbf{p}} = \mathbf{1} - \mathbf{u}$, where $\mathbf{u}$ is the leading left singular vector of $Y - \mathbf{1}\mathbf{1}^T$. The resulting overlap estimates $\hat{p}_1, \ldots, \hat{p}_n$ are called the *Spectral Jaccard Similarity* scores and were shown to provide a much better estimate of overlap sizes than the estimator given by (3), by accounting for the variable quality of hash functions for the task.

In this paper, motivated by the model of Baharav et al. [5], we consider the more general framework of rank-one models. In this setting, a vector of parameters $\mathbf{u} = [u_1, \ldots, u_n]^T$ (the item qualities) is to be estimated from the binary responses provided by $m$ workers, and the $n \times m$ observation matrix $X$ is assumed to satisfy $\mathbb{E}X = \mathbf{u}\mathbf{v}^T$. In the context of these rank-one models, a natural estimator for $\mathbf{u}$ is the leading left singular vector of $X$. Such a spectral estimator has been shown to have good performance both in the context of pairwise sequence alignment [5] and in voting aggregation applications [19, 26]. However, the spectral decomposition by default allocates worker resources uniformly across all items. In practice, one is often only interested in identifying the "most popular" items, which, in the context of pairwise sequence alignment, corresponds to the reads $S_i$ that have the largest overlaps with a reference read $S_0$. Hence, we seek strategies that can harness the performance of spectral methods while using adaptivity to avoid wasting worker resources on unpopular items.

**Main contributions:** We propose an adaptive spectral estimation algorithm, based on multi-armed bandits, for identifying the $k$ largest entries of the leading left singular vector $\mathbf{u}$ of $\mathbb{E}X$. A key

technical challenge is that multi-armed bandit algorithms generally rely on our ability to build confidence intervals for each arm, but it is difficult to obtain tight *element-wise* confidence intervals for the singular vectors of random matrices with low expected rank [1]. For that reason, we propose a variation of the spectral estimator for $\mathbf{u}$, in which one computes the leading *right* singular vector first and uses it to estimate the entries of $\mathbf{u}$ via a *matched filter* [44]. This allows us to compute entrywise confidence intervals for each $u_i$, which in turns allows us to adapt the Sequential Halving bandit algorithm of Karnin et al. [28] to identify the top-$k$ entries of $\mathbf{u}$. We provide theoretical performance bounds on the total workers' response budget required to correctly identify the top-$k$ items with a given probability. We empirically validate our algorithm on controlled experiments that simulate the vote aggregation scenario and on real data in the context of pairwise alignment of DNA sequences. For a PacBio *E. coli* dataset [38], we show that adaptivity can reduce the budget requirements (which correspond to the number of min-hash comparisons) by around half.

**Related work:** Our work is motivated by the bioinformatics literature on pairwise sequence alignment [6, 29, 30, 36]. In particular, we build on the idea of using min-hash-based techniques to efficiently estimate pairwise sequence alignments [6], described by the estimator in (3). More sophisticated versions of this idea have been proposed, such as the use of a *tf-idf* (term frequency-inverse document frequency) weighting for the different hash functions [11, 34], which follows the same observation that "some hashes are better than others" made by Baharav et al. [5]. A proposed strategy to reduce the number of hash functions needed for the alignment estimates is to use *bottom sketches* [37]. More precisely, for a single hash function, one can compute $s$ minimisers per read (the bottom-$s$ sketch) and estimate the alignments based on the size of the intersection of bottom sketches. *Winnowing* has also been used in combination with min-hash techniques to allow the mapping of reads to very long sequences, such as full genomes [23].

The literature on crowdsourcing and vote aggregation is vast [14, 19, 26, 27, 31, 40, 41, 46, 47, 47–50]. Many of these works are motivated by the classical Dawid-Skene model [16]. Ghosh et al. [19] considered a setting where the questions have binary answers and the workers' responses are noisy. They proposed a rank-one model and used a spectral method to estimate the true answers. For a similar setting, Karger et al. [27] showed that random allocation of questions to workers (according to a sparse random graph) followed by belief propagation is optimal to obtain answers to the questions with some probability of error, and this work was later extended [14, 26]. While the rank-one model these works have considered is similar in spirit to the one we consider, in our setting, the *true* answers to the questions are not binary. Rather, our answers are parameters in $(0, 1)$, which can be thought of as the quality of an item or the fraction of the population that would answer "yes" to a question. Natural tasks to consider in our case would be to find the top ten products among a catalogue of 100,000 products, or identify the items that are liked by more than 80% of the population. Notice that such questions are meaningless in the case of questions with binary answers.

The use of adaptive strategies for ranking a set of items or identifying the top-$k$ items have been studied in the context of *pairwise comparisons* between the items [8, 20, 21, 42]. Moreover, the top-$k$ arm selection problem has been considered in many contexts and a large set of algorithms exist to identify the best items while minimising the number of observations required to do that [7, 17, 22, 24, 25, 28, 35]. Recent works have taken advantage of some of these multi-armed bandit algorithms to solve large-scale computation problems [2–4], which is similar in flavor to our work.

**Outline of manuscript:** In Section 2, we introduce rank-one models and give examples of potential applications. In Section 3, we develop a spectral estimator for these models that enables construction of confidence intervals. In Section 4, we leverage these confidence intervals to develop adaptive bandit-based algorithms. Section 5 presents empirical results.

## 2   Rank-One Models

While our main target application is the pairwise sequence alignment problem, we define rank-one models for the general setting of response aggregation problems. In this setting, we are interested in estimating a set of parameters, or item values, $u_1, \ldots, u_n$. To do that, we recruit a set of workers with unknown levels of expertise, which provide binary opinions about the items. We can choose the number of workers and their opinions can be requested for any subset of the $n$ items. The rank-one model assumption is that the matrix of responses $X = [X_{i,j}]$ is rank-one in expectation; i.e.,

$$\mathbb{E}X = \mathbf{u}\mathbf{v}^T \tag{6}$$

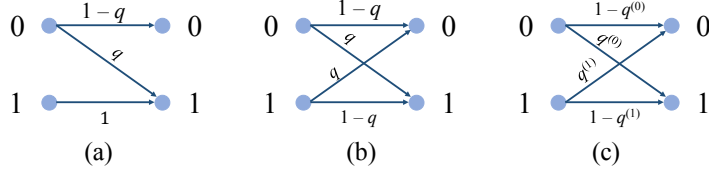

Figure 1: *Three Channel Models*: (a) the $Z$-channel models one-sided errors; (b) the binary symmetric channel models two-sided symmetric errors; (c) the general binary channel admits different probabilities $q^{(0)}$ for $0 \to 1$ errors and $q^{(1)}$ for $1 \to 0$ errors.

for unknown vectors $\mathbf{u}$ and $\mathbf{v}$ with entries in $(0,1)$. This means that, for each $i$ and $j$, $X_{i,j}$ has a $\mathrm{Ber}(u_i v_j)$ distribution. Furthermore, we assume throughout that all entries $X_{i,j}$ are independent.

Abstractly, this setting applies to a situation in which we have a list of questions that we want answered. One can think of a question as the rating of a movie or whether a product is liked. The parameter associated with the $i$th question, $u_i$, can be thought of as representing the average rating of the movie, or the fraction of people in the population who like the product. We can think of $X_{i,j}$ as a noisy version of the response of worker $j$ to question $i$, with the noise modelling respondent error. We point out that $X$ can alternatively be viewed as noisy observations of responses to the questions through a *binary channel* [13], with each worker having a channel of distinct characteristics. We next discuss two instances of this model and the corresponding binary channels.

**One-sided error:** In the pairwise sequence alignment problem presented in Section 1, we want to recover the pairwise overlap fractions (or alignment scores) $p_i \in (0,1)$ between read $S_0$ and read $S_i$, for $i = 1, \dots, n$. The observation for read pair $(S_0, S_i)$ and hash function $h_j$ is modelled by $Y_{i,j} \sim \mathrm{Ber}(p_i) \vee \mathrm{Ber}(q_j)$, which can be thought of as observing a $\mathrm{Ber}(p_i)$ random variable through a $Z$-*channel* with crossover probability $q_j$. A Z-channel, shown in Fig. 1(a), is one where an input $1$ is never flipped but an input $0$ is flipped with some probability [13]. Hence, this models one-sided errors. If we process the data as $X \triangleq Y - \mathbf{1}\mathbf{1}^T$, we then have that

$$\mathbb{E}[X] = (\mathbf{1} - \mathbf{p})(\mathbf{q} - \mathbf{1})^T, \tag{7}$$

giving us $\mathbf{u} = \mathbf{1} - \mathbf{p}$ and $\mathbf{v} = \mathbf{1} - \mathbf{q}$. While, in this case, the entries $X_{i,j}$ are technically in $\{0, -1\}$, our main results (presented in Section 3 for a binary matrix $X \in \{0,1\}^{n \times m}$) can be readily extended.

**Two-sided error:** In the binary crowdsourcing problem, $n$ items have associated parameters $p_i \in (0,1)$ (the population rating of the item), for $i = 1, \dots, n$. The observation of the rating of worker $j$ on the $i$th item is modelled as $Y_{i,j} \sim \mathrm{Ber}(p_i) \oplus \mathrm{Ber}(q_j)$, where $\oplus$ represents the XOR operation. This can be thought of as observing a $\mathrm{Ber}(p_i)$ random variable through a *Binary Symmetric Channel* with crossover probability $q_j$. A Binary Symmetric Channel, shown in Fig. 1(b), is one where the probability of flipping 1 to 0 and 0 to 1 is the same. The processed data in this case is $X \triangleq Y - \frac{1}{2}\mathbf{1}\mathbf{1}^T$, and the expected value of our observation matrix is given by

$$\mathbb{E}[X] = \left(\mathbf{p} - \tfrac{1}{2}\mathbf{1}\right)\left(\mathbf{1} - 2\mathbf{q}\right)^T, \tag{8}$$

giving us $\mathbf{u} = \mathbf{p} - \frac{1}{2}\mathbf{1}$ and $\mathbf{v} = \mathbf{1} - 2\mathbf{q}$. As in the case of one-sided errors, the observations $X_{i,j}$ are not in $\{0,1\}$ but they only take two values and the results in Section 3 still hold.

Notice that if we do not make the assumption of symmetry and use a general binary channel, shown in Fig. 1(c), the model is still low-rank (it is rank-2) in expectation. In this manuscript, we focus on rank-one models, but briefly discuss this generalization in Appendix A.

**Model Identifiability:** Strictly speaking, the models described above are not identifiable, unless extra information is provided. This is because $\|\mathbf{u}\|$ and $\|\mathbf{v}\|$ are unspecified, and replacing $\mathbf{u}$ and $\mathbf{v}$ with $\alpha \mathbf{u}$ and $\frac{1}{\alpha}\mathbf{v}$ leads to the same distribution for the observation matrix $X$.

In practice one can overcome this issue by including questions with known answers. For the pairwise sequence alignment problem of Baharav et al. [5], the authors add "calibration reads" to the dataset. These are random reads, expected to have zero overlap with other reads in the dataset. In a crowdsourcing setting one similarly can add questions whose answers are known to the dataset. Based on questions with known answers, it is possible accurately estimate the "average expertise" of the workers, captured by $\|\mathbf{v}\|$. In order to avoid overcomplicating the model and the results and to

circumvent the unidentifiability issue, we assume that $\|\mathbf{v}\|$ is known in our theoretical analysis. In our experiments, we adopt the known-questions strategy to estimate $\|\mathbf{v}\|$.

# 3 Spectral Estimators

Consider the general rank-one model described in Section 2. The $n \times m$ binary matrix of observations $X$ can be written as $X = \mathbb{E}[X] + W$, where $\mathbb{E}[X] = \mathbf{u}\mathbf{v}^T$. We assume throughout this section that $u_i, v_j \in [c, C]$, where $0 < c < C < 1$. Moreover, we assume that $\|\mathbf{v}\|$ is known, in order to make the model identifiable, as described in Section 2.

A natural estimator for $\mathbf{u}$ is the leading left singular vector of $X$ (or the leading eigenvector of $XX^T$), rescaled to have the same norm as $\mathbf{u}$. One issue with such an estimator is that it is not straightforward to obtain confidence intervals for each of its entries. There is a fair amount of work in constructing $\ell_2$ confidence intervals around eigenvectors of perturbed matrices [9, 10, 12]. However, the translation of $\ell_2$ control over eigenvectors to element-wise control using standard bounds costs us a $\sqrt{n}$ factor, which makes the resulting bounds too loose for the purposes of adaptive algorithms, which we will explore in Section 4. There has been some work on directly obtaining $\ell_\infty$ control for eigenvectors by Abbe et al. [1], Fan et al. [18]. However, a direct application of these results to our rank-one models does not give us enough element-wise control for our purposes.

In order to overcome this issue and obtain element-wise confidence bounds on our estimate of each $u_i$, we propose a variation on the standard spectral estimator for $\mathbf{u}$. To provide intuition to our method, let us consider a simpler setting – one where we know $\mathbf{v}$ exactly. In this case a popular means to estimate $\mathbf{u}$ is the *matched filter* estimator [44]

$$\hat{u}_i = X_{i,\cdot} \frac{\mathbf{v}}{\|\mathbf{v}\|^2}, \tag{9}$$

where $X_{i,\cdot}$ is the $i$th row of $X$. It is easy to see that $\hat{u}_i$ is an unbiased estimator of $u_i$, and standard concentration inequalities can be used to obtain confidence intervals. We try to mimic this intuition by splitting the rows of the matrix into two – red rows and blue rows. We then use the red rows to obtain an estimate $\hat{\mathbf{v}}$ of $\mathbf{v}$. We treat this as the true value of $\mathbf{v}$ and obtain the matched filter estimate for the $u_i$s corresponding to the blue rows, which gives us element-wise confidence intervals. We then use the blue rows to estimate $\mathbf{v}$, and apply the matched filter to obtain estimates for the $u_i$s corresponding to the red rows. This is summarised in the following algorithm.

---
**Algorithm 1** Spectral estimation of $\mathbf{u}$

---
1: **Input:** $X \in \{0, 1\}^{n \times m}$, $\|\mathbf{v}\|$
2: Split $X$ into two $\frac{n}{2} \times m$ matrices $X_A$ and $X_B$
3: $\hat{\mathbf{v}}_A \leftarrow$ leading right singular vector of $X_A$
4: $\hat{\mathbf{v}}_B \leftarrow$ leading right singular vector of $X_B$
5: $\hat{\mathbf{u}}_A \leftarrow X_A \frac{\hat{\mathbf{v}}_B}{\|\hat{\mathbf{v}}_B\|\|\mathbf{v}\|}$, $\hat{\mathbf{u}}_B \leftarrow X_B \frac{\hat{\mathbf{v}}_A}{\|\hat{\mathbf{v}}_A\|\|\mathbf{v}\|}$
6: **return** $\hat{\mathbf{u}} = \begin{bmatrix} \hat{\mathbf{u}}_A \\ \hat{\mathbf{u}}_B \end{bmatrix}$

---

The main result in this section is an element-wise confidence interval for the resulting $\hat{\mathbf{u}}$.

**Theorem 1.** *When given $X$ and $\|\mathbf{v}\|$ as inputs, Algorithm 1 returns $\hat{\mathbf{u}} = [\hat{u}_1, \ldots, \hat{u}_m]^T$ satisfying*

$$\mathbb{P}\left(|\hat{u}_i - u_i| > \epsilon\right) \le 3n \exp\left(-C_1 m \epsilon^2\right), \tag{10}$$

*for $i \in \{1, \ldots, n\}$, $0 < \epsilon < 1$, $m \le n$, and constant $C_1$ specified in Appendix G.*

In the remainder of this section, we describe the key technical results required to prove Theorem 1. We discuss the application of these confidence intervals to create an adaptive algorithm in Section 4.

To prove Theorem 1 we first establish a connection between $\ell_2$ control of $\hat{\mathbf{v}}$ and element-wise control of $\hat{\mathbf{u}}$. Then we provide expectation and tail bounds for the $\ell_2$ error in $\hat{\mathbf{v}}$. For ease of exposition, we will drop the subscripts $A$ and $B$ in $X_A, X_B, \hat{\mathbf{v}}_A$, and $\hat{\mathbf{v}}_B$. We will implicitly assume that $X$ and $\hat{\mathbf{v}}$ correspond to distinct halves of the data matrix, thus being independent. The main technical ingredient required to establish Theorem 1 is the following lemma.

**Lemma 1.** *The error of estimator $\hat{\mathbf{u}}$ satisfies*

$$\|\hat{\mathbf{u}} - \mathbf{u}\|_\infty \leq \frac{1}{\|\mathbf{v}\|}\left|\sum_{j=1}^m (X_{i,j} - u_i)\,\mathbb{E}\left[\frac{\hat{v}_j}{\|\hat{\mathbf{v}}\|}\right]\right| + \frac{1}{c}\left\|\frac{\hat{\mathbf{v}}}{\|\hat{\mathbf{v}}\|} - \frac{\mathbf{v}}{\|\mathbf{v}\|}\right\| + \left(1 + \frac{1}{c}\right)\mathbb{E}\left\|\frac{\hat{\mathbf{v}}}{\|\hat{\mathbf{v}}\|} - \frac{\mathbf{v}}{\|\mathbf{v}\|}\right\|.$$

Notice that the right-hand side of the bound in Lemma 1 (proved in Appendix B) involves the $\ell_2$ error of $\hat{\mathbf{v}}$, and can in turn be used to bound the $\ell_\infty$ error of the estimator $\hat{\mathbf{u}}$. The first term in the bound is a sum of independent, bounded, zero-mean random variables $(X_{i,j} - u_i)\mathbb{E}[\hat{v}_j/\|\hat{\mathbf{v}}\|]$, for $j = 1, \ldots, m$. Using Hoeffding's inequality, we show in Appendix C that, for any $\epsilon > 0$,

$$\mathbb{P}\left(\left|\sum_{j=1}^m (X_{i,j} - u_i)\mathbb{E}[\hat{v}_j/\|\hat{\mathbf{v}}\|]\right| > \|\mathbf{v}\|\epsilon\right) \leq 2\exp\left(-2c^2 m\epsilon^2\right). \tag{11}$$

In order to bound the second and third terms on the right-hand side of Lemma 1, we resort to matrix concentration inequalities and the Davis-Kahan theorem [15]. More precisely, we have the following lemma, which we prove in Appendix D.

**Lemma 2.** *The error of estimator $\hat{\mathbf{v}}$ satisfies*

*(a)* $\mathbb{P}\left(\left\|\frac{\hat{\mathbf{v}}}{\|\hat{\mathbf{v}}\|} - \frac{\mathbf{v}}{\|\mathbf{v}\|}\right\| \geq \epsilon\right) \leq (m+n)\exp\left(-C_2\epsilon^2\min(m,n)\right), \quad for\ 0 < \epsilon < 1,$

*(b)* $\mathbb{E}\left\|\frac{\hat{\mathbf{v}}}{\|\hat{\mathbf{v}}\|} - \frac{\mathbf{v}}{\|\mathbf{v}\|}\right\| \leq C_3\sqrt{\frac{\log(m+n)}{\min(m,n)}},$

*where $C_2$ and $C_3$ are constants specified in Appendix G.*

Given (11) and the bounds in Lemma 2, it is straightforward to establish Theorem 1, as we do next. Combining Lemmas 1, 2 and equation (11) to build a confidence interval for $\hat{u}_i$. To that end, fix some $\epsilon \in (0,1)$. If $\epsilon^2\min(m,n) > (6C_3/c)^2\log(m+n)$, Lemma 2(b) implies that

$$\left(1 + \frac{1}{c}\right)\mathbb{E}\left\|\frac{\hat{\mathbf{v}}}{\|\hat{\mathbf{v}}\|} - \frac{\mathbf{v}}{\|\mathbf{v}\|}\right\| \leq \frac{2C_3}{c}\sqrt{\frac{\log(m+n)}{\min(m,n)}} < \frac{\epsilon}{3}.$$

Hence, if $\epsilon^2\min(m,n) > (6C_3/c)^2\log(m+n)$,

$$\mathbb{P}\left(|\hat{u}_i - u_i| > \epsilon\right) \leq \mathbb{P}\left(\left|\sum_{j=1}^m (X_{i,j} - p_i)\mathbb{E}\left[\frac{\hat{v}_j}{\|\hat{\mathbf{v}}\|}\right]\right| > \frac{\|\mathbf{v}\|\epsilon}{3}\right) + \mathbb{P}\left(\left\|\frac{\hat{\mathbf{v}}}{\|\hat{\mathbf{v}}\|} - \frac{\mathbf{v}}{\|\mathbf{v}\|}\right\| \geq \frac{c\epsilon}{3}\right)$$

$$\leq 2\exp\left(-\frac{c^2 m\epsilon^2}{18}\right) + (m+n)\exp\left(-C_2\frac{c^2\epsilon^2\min(m,n)}{9}\right)$$

$$\leq (m+n+2)\exp\left(-C_4\min(m,n)\epsilon^2\right). \tag{12}$$

Notice that (12) is a vacuous statement whenever $C_4\epsilon^2\min(m,n) < \log(m+n)$, as the right-hand side is greater than 1. Hence, if we replace $C_4$ with $C_1 = \min[C_4, (6C_3/c)^{-2}]$, the inequality holds for all $m$ and $n$. The result in Theorem 1 then follows by assuming $m \leq n$.

## 4  Leveraging confidence intervals for adaptivity

In the pairwise sequence alignment problem, one is typically only interested in identifying pairs of reads with large overlaps. Hence, by discarding pairs of reads with a small overlap based on a coarse alignment estimate and adaptively picking the pairs of reads for which a more accurate estimate is needed, it is possible to save significant computational resources. Similarly, in crowdsourcing applications, one may be interested in employing adaptive schemes in order to effectively use worker resources to identify only the most popular items.

We consider two natural problems that can be addressed within an adaptive framework. The first one is the identification of the top-$k$ largest alignment scores. In the second problem, the goal is to return a list of reads with high pairwise alignment to the reference, i.e., all reads with $u_i$ above a certain threshold. More generally, we consider the task of identifying a set of reads including *all* reads

with pairwise alignment above $\alpha$ and no reads with pairwise alignment below $\beta$, for some $\beta \leq \alpha$. Adaptivity in the first problem can be achieved by casting the problem as a top-$k$ arm multi-armed bandit problem, while the second problem can be cast as a Thresholding Bandit problem [32].

**Identifying the top-$k$ alignments:** We consider the setting where we wish to find the $k$ largest pairwise alignments with a given read. We assume that we have a total computational budget of $T$ min-hash comparisons. Notice that the regime where $T < n$ is uninteresting as we cannot even make one min-hash comparison per read. When $T = \Omega(n)$, a simple non-adaptive approach is to divide our budget $T$ evenly among all reads (as done in [5]). This gives us an $n \times \frac{T}{n}$ min-hash collision matrix $X$, from which we can estimate $\hat{\mathbf{u}}$ using Algorithm 1 and choose the top $k$ alignments based on their $\hat{u}_i$ values. Let $u_{(1)} \geq u_{(2)} \geq \ldots \geq u_{(n)}$ be the sorted entries of the true $\mathbf{u}$ and define $\Delta_i = u_{(i)} - u_{(k)}$ for $i = 1, \ldots, n$. Notice that the non-adaptive approach recovers $u_{(1)}, \ldots, u_{(k)}$ correctly if each $\hat{u}_i$ stays within $\Delta_{k+1}$ of its true value $u_i$. From the union bound and Theorem 1,

$$\mathbb{P}(\text{failure}) \leq \sum_{i=1}^{n} \mathbb{P}\left(|\hat{u}_i - u_i| > \Delta_{k+1}/2\right) \leq 3n^2 \exp\left(-\frac{C_1}{4} \frac{\Delta_{k+1}^2 T}{n}\right), \quad (13)$$

if $n \leq T \leq n^2$. Hence, the budget required to achieve an error probability of $\delta$ is

$$T = O\left(\Delta_{k+1}^{-2} n \log\left(\frac{n}{\delta}\right)\right). \quad (14)$$

Moreover, from (13), we see that a budget $T = n \log^{\beta} n$, $\beta > 1$, allows you to correctly identify the top-$k$ alignments if $(\Delta_{k+1}^2 T)/n \approx \log n$, or $\Delta_{k+1} \approx \log^{-(\beta-1)/2} n$. Hence, the budget $T$ places a constraint in the minimum gap $\Delta_{k+1}$ that can be resolved.

Next we propose an adaptive way to allocate the same budget $T$. Algorithm 2 builds on an approach by Karnin et al. [28], but incorporates the spectral estimation approach from Section 3. The algorithm assumes the regime $n \log n < T < n^2$.

---

**Algorithm 2** Adaptive Spectral Top-$k$ Algorithm

---

1: **Input:** $T$, $k$
2: Initialize $\mathcal{I}_0 \leftarrow \{1, 2, \ldots, n\}$ &emsp;&emsp;&emsp;&emsp;&emsp;&emsp;&emsp;&emsp;&emsp; $\triangleright$ Initial set of candidates
3: **for** $r = 0$ **to** $r_{\max} \triangleq \lceil \log_2 \frac{n}{\sqrt{T}} \rceil - 1$ **do**
4: &emsp;&emsp; $t_r \leftarrow \left\lfloor \dfrac{T}{2|\mathcal{I}_r| \lceil \log_2 \frac{n}{\sqrt{T}} \rceil} \right\rfloor$ &emsp;&emsp;&emsp;&emsp;&emsp;&emsp; $\triangleright$ Number of samples to be taken
5: &emsp;&emsp; Obtain a binary matrix $X^{(r)} \in \{0,1\}^{|\mathcal{I}_r| \times t_r}$ and corresponding $\|\mathbf{v}^{(r)}\|$
6: &emsp;&emsp; Use Algorithm 1 to compute estimates $\hat{\mathbf{u}}^{(r)}$ for $X^{(r)}$
7: &emsp;&emsp; Set $\mathcal{I}_{r+1}$ to be the $\lceil |\mathcal{I}_r|/2 \rceil$ coordinates in $\mathcal{I}_r$ with largest $\hat{\mathbf{u}}^{(r)}$
8: **Clean up:** Use $t_{r_{\max}+1} = \frac{T}{2}$, and compute $\hat{\mathbf{u}}^{(r_{\max}+1)}$ as above
9: **return** the $k$ coordinates of $\mathcal{I}_{r_{\max}+1}$ with the largest $\hat{\mathbf{u}}^{(r_{\max}+1)}$

---

At the $r$-th iteration, Algorithm 2 uses $t_r$ new hash functions and computes the min-hash collisions for the reads in $\mathcal{I}_r$, which are represented by the matrix $X^{(r)}$. Notice that we assume that the $t_r$ min-hashes in each iteration are different, which makes observation matrices $X^{(r)}$ all independent. Also, we assume that the $\ell_2$ norm of the right singular vector of $\mathbb{E}X^{(r)}$, $\|\mathbf{v}^{(r)}\|$, can be obtained exactly at each iteration of the algorithm. As discussed in Section 2, this makes the model identifiable and can be emulated in practice with calibration reads. At each iteration, Algorithm 2 eliminates half of the reads in $\mathcal{I}_r$. After $r_{\max} + 1$ iterations, the number of remaining reads satisfies $\sqrt{T}/2 \leq |\mathcal{I}_{r_{\max}+1}| \leq 2\sqrt{T}$, and the total budget used is $\sum_{r=0}^{r_{\max}} |\mathcal{I}_r| t_r \leq T/2$. Finally, in the "clean up" stage, we use the remaining $T/2$ budget to obtain the top $k$ among the approximately $\sqrt{T}$ remaining items. Notice that the final observation matrix is approximately $\sqrt{T} \times \sqrt{T}$.

In order to analyse the performance of Algorithm 2, we follow Karnin et al. [28], and define $H_2 = \max_{i>k} \frac{i}{\Delta_i^2}$. We then have the following performance guarantee (proof in Appendix F).

**Theorem 2.** *Given budget* $n \log n \leq T \leq n^2$ *and assuming* $2k < \sqrt{T}$, *Algorithm 2 correctly identifies the top* $k$ *alignments with probability at least*

$$1 - 18kn \log n \exp\left(-\frac{C_1}{64} \frac{T}{H_2 \log n}\right) - 12n^2 \exp\left(-\frac{C_1}{16} \Delta_{k+1}^2 \sqrt{T}\right) \tag{15}$$

*Moreover, for sufficiently small* $\delta$, *Algorithm 2 achieves an error probability of at most* $\delta$ *with budget*

$$T = O\left(H_2 \log n \log\left(nk\frac{\log n}{\delta}\right) + \Delta_{k+1}^{-4} \log^2\left(\frac{n^2}{\delta}\right)\right). \tag{16}$$

Comparing (15) and (16) with the non-adaptive counterparts (13) and (14) requires a handle on $H_2$. This quantity captures how difficult it is to separate the top $k$ alignments, and satisfies $(k+1)\Delta_{k+1}^{-2} \leq H_2 \leq n\Delta_{k+1}^{-2}$. The extreme case $H_2 \approx n\Delta_{k+1}^{-2}$ occurs when all of the $n-k$ suboptimal items have very similar qualities. In this case, adaptivity is not helpful. However, when $u_{(k+1)}$ is large compared to other non-top-$k$ alignments, we are in the $H_2 \approx k\Delta_{k+1}^{-2}$ regime. Then the budget requirements are essentially $T = O(k\Delta_{k+1}^{-2} \log^2 n + \Delta_{k+1}^{-4} \log^2 n)$, which is $O(\Delta_{k+1}^{-4} \log^2 n)$ for $k, \delta$ constant. Furthermore, in the $H_2 \approx k\Delta_{k+1}^{-2}$ regime, a budget $T = n \log^\beta n$, $\beta > 2$, allows you to correctly identify the top-$k$ alignments with a gap of $\Delta_{k+1} \approx n^{-1/4}$ which is significantly smaller than the $\Delta_{k+1} \approx \log^{-(\beta-1)/2} n$ afforded in the non-adaptive case. As a concrete example, suppose that out of the $n$ alignments, there are $k$ highly overlapping reads with $u_i = C$, $k$ moderately overlapping reads with $u_i = C - n^{-1/4}$, and $n - 2k$ reads with no overlap and $u_i = c$. In this case, Algorithm 2 requires a budget of $T = O\left(n \log^2\left(\frac{n}{\delta}\right)\right)$, while the non-adaptive approach requires $T = O\left((n^{3/2} \log\left(\frac{n}{\delta}\right)\right)$.

**Identifying all alignments above a threshold:** In this setting, we wish to select a set of reads such that with high probability all reads with an overlap $u_i \geq \beta$ with the reference read are returned and no reads with $u_i \leq \alpha$ are returned, for some $\beta > \alpha$. We state the following theorem, but relegate the discussion and presentation of Algorithm 3 to Appendix E.

**Theorem 3.** *Given parameters* $\beta$ *and* $\alpha$ *such that* $\beta - \alpha > \sqrt{\frac{12 \log n}{C_4 n}}$, *with probability at least* $1 - \frac{2}{n}$ *Algorithm 3 will output a set of reads* $R$ *such that* $\{i : u_i > \beta\} \subseteq R \subseteq \{i : u_i > \alpha\}$ *and use budget*

$$T \leq 2\left(\frac{12}{C_4} \frac{\log n}{(\beta-\alpha)^2}\right)^2 + \sum_{\ell=\kappa+1}^{n} \frac{32}{C_4} \frac{\log n}{(\Gamma^{(\ell)})^2} \quad for \quad \Gamma_i = \begin{cases} u_i - \alpha & if \ \beta < u_i, \\ \beta - \alpha & if \ \alpha \leq u_i \leq \beta, \\ \beta - u_i & if \ u_i < \alpha, \end{cases} \tag{17}$$

*where* $\Gamma_i$ *denotes the difficulty of classifying* $u_i$, *with* $\Gamma^{(1)} \leq \cdots \leq \Gamma^{(n)}$ *as the sorted list of the* $\Gamma_i$.

## 5 Empirical Results

In order to validate the Adaptive Spectral Top-$k$ algorithm, we conducted two types of experiments: (1) controlled experiments on simulated data for a crowdsourcing model with symmetric errors; (2) pairwise sequence alignment experiments on real DNA sequencing data. We consider the top-$k$ identification problem with $k = 5$ in both cases. We run Algorithm 2 with some slight modifications, namely halving until we have fewer than $2k$ remaining arms before moving to the clean up step, and compare its performance with the non-adaptive spectral approach. Further experimental details are in Appendix I. We measure success in two ways. First, we consider the error probability of returning the top $k$ items (i.e., any deviation from the top-$k$ is considered a failure). Second, we consider a less stringent metric, where we allow the algorithm to return its top-$2k$ items, and we consider the fraction of the true top-$k$ items that are present to evaluate performance. Our code is publicly available online at github.com/TavorB/adaptiveSpectral.

**Controlled experiments:** We consider a crowdsourcing scenario with symmetric errors as modelled in (8). We want to determine the 5 best products from a list of 1000 products. We generate the true product qualities (that is, the $p_i$ parameters) from a Beta$(1, 5)$ distribution independent of each other. Each of the worker abilities $q_j$ is drawn from a Uniform$(0, 1)$ distribution, independent of everything else. We consider the problem of top-5 product detection at various budgets as shown in Figure 2(a) with success rate measured by the presence in the top-10 items. We see that the adaptive algorithm requires significantly fewer worker responses to achieve equal performance to the non-adaptive one.

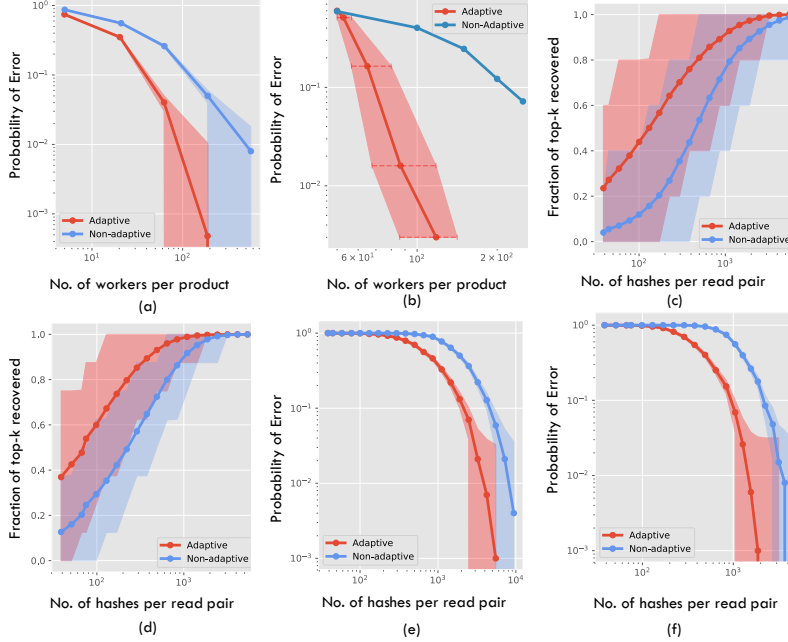

Figure 2: (a) shows the probability of error of the controlled crowdsourcing experiment as the number of workers per product is increased, where an error is defined as incorrectly identifying the set of top-$k$ products. (b) shows the same in the thresholding bandits setup. (c) shows the fraction of the top-$k$ reads that are in the $2k$ reads returned for the *E. coli* dataset, while (d) shows the fraction of the top-$k$ reads that are in the top-$k$ reads returned for the NCTC4174 dataset. (e) shows the probability of error of correctly identifying the set of top-$k$ overlaps on the *E. coli* dataset, while (f) shows the same for the NCTC4174 dataset. 1000 trials are conducted for each point. $95\%$ percentiles are shaded around each point in (b), (c) and (d) (note that confidence intervals for (b) are on the x-axis). For (a), (e) and (f), $\frac{1}{\sqrt{\text{number of trials}}}$ is shaded around each point. For further details, see Appendix I.

In Figure 2(b) we consider the same set up as above but in the fixed confidence setting, considering the problem of being able to detect all products that are liked by more than $65\%$ of the population while returning none that is liked by less than $50\%$ of the population. Again, we see that for the same probability of error the adaptive algorithm needs far fewer workers.

**Real data experiments:** Using the PacBio *E. coli* data set [38] that was examined in Baharav et al. [5] we consider the problem of finding, for a fixed reference read, the $5$ reads that have the largest alignment with the reference read in the dataset. We show the fraction of the $5$ reads that are present when returning $10$ reads in Figure 2(c) and the success probability when returning exactly $5$ reads in Figure 2(e) (i.e., the probability of returning exactly the top-5 reads). To achieve an error rate of $0.9\%$ the non-adaptive algorithm requires over $8500$ min-hash comparisons per read, while the adaptive algorithm requires fewer than $6000$ per read to reach an error rate of $0.1\%$.

We also consider the NCTC4174 dataset of [39] and plot the fraction correct when returning $5$ reads in Figure 2(d) and the success probability when returning exactly $5$ reads in Figure 2(f). The results are qualitatively similar to what we observe in the case of the *E. coli* dataset.

## 6 Discussion

Motivated by applications in sequence alignment, we considered the problem of efficiently finding the largest elements in the left singular vector of a binary matrix $X$ with $\mathbb{E}[X] = \mathbf{u}\mathbf{v}^T$. To utilize the natural spectral estimator of $\mathbf{u}$, we designed a method to construct $\ell_\infty$ confidence intervals around the spectral estimator. To perform this spectral estimation efficiently, we leveraged multi-armed bandit algorithms to adaptively estimate the entries $u_i$ of the leading left singular vector to the necessary degree of accuracy. We show that this method provides computational gains on both real data and in controlled experiments.

## Broader Impact

Over the last decade, high-throughput sequencing technologies have driven down the time and cost of acquiring biological data tremendously. This has caused an explosion in the amount of available genomic data, allowing scientists to obtain quantitative insights into the biology of all living organisms. Countless tasks – such as gene expression quantification, metagenomic sequencing, and single-cell RNA sequencing – heavily rely on some form of pairwise sequence alignment, which is a heavy computational burden and often the bottleneck of the analysis pipeline. The development of efficient algorithms for this task, which is the main outcome of this paper, is thus critical for the scalability of genomic data analysis.

From a theoretical perspective, this work establishes novel connections between a classical problem in bioinformatics (pairwise sequence alignment), spectral methods for parameter estimation from crowdsourced noisy data, and multi-armed bandits. This will help facilitate the transfer of insights and algorithms between these traditionally disparate areas. It will also add a new set of techniques to the toolbox of the computational biology community that we believe will find a host of applications in the context of genomics and other large-scale omics data analysis. Further, this work will allow other Machine Learning researchers unfamiliar with bioinformatics to utilise their expertise in solving new problems at this novel intersection of bioinformatics, spectral methods, and multi-armed bandits.

## Acknowledgments and Disclosure of Funding

G.M. Kamath would like to thank Lester Mackey of Microsoft Research, New England Lab for useful discussions on Rank-one models and their connection to crowd-sourcing. The research of T. Baharav was supported by the Alcatel Lucent Stanford Graduate Fellowship and NSF GRFP. The research of I. Shomorony was supported in part by NSF grant CCF-2007597.

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
