[Supplementary Material]

# Appendices

## A  A Rank-$2$ Model

Consider the case where we the $(i, j)$-th observation $Y_{i,j}$ is the output of a $\text{Ber}(p_i)$ random variable passed through a general binary channel $\text{BC}(q_j^{(0)}, q_j^{(1)})$ (see Figure 1(c)). Here $q_j^{(0)}$ is probability of a $0$ being flipped to a $1$ and $q_j^{(1)}$ is probability of a $1$ being flipped to a $0$ on the $j$th column.

We note that we have $n + 2m$ parameters here, while a rank-one model would admit only $n + m$ parameters. Hence this is not a rank-one model. However we note that

$$\mathbb{E}[Y] = \mathbf{p}(\mathbf{1} - \mathbf{q}^{(0)})^T + (\mathbf{1} - \mathbf{p})\mathbf{q}^{(1)T}, \tag{18}$$

where $\mathbf{p} = [p_1, \ldots, p_n]^T$, $\mathbf{q}^{(0)} = [q_1^{(0)}, \ldots, q_m^{(0)}]^T$ and $\mathbf{q}^{(1)} = [q_1^{(1)}, \ldots, q_m^{(1)}]^T$. This shows that, when the noise in the workers' responses is modelled by a general binary channel (see Figure 1(c)), we have a rank-2 model.

## B  Proof of Lemma 1

We start by using the triangle inequality to obtain

$$|\hat{u}_i - u_i| \leq |\hat{u}_i - \mathbb{E}[\hat{u}_i]| + |\mathbb{E}[\hat{u}_i] - u_i|. \tag{19}$$

To bound the first term, we first notice that since

$$\hat{u}_i = X_{i,\cdot} \frac{\hat{\mathbf{v}}}{\|\hat{\mathbf{v}}\|\|\mathbf{v}\|} = \frac{1}{\|\mathbf{v}\|} \sum_{j=1}^{m} \left( X_{i,j} \frac{\hat{v}_j}{\|\hat{\mathbf{v}}\|} \right) \tag{20}$$

and $\hat{\mathbf{v}}$ is independent of $X_{i,\cdot}$, we have that

$$
\begin{aligned}
\hat{u}_i - \mathbb{E}[\hat{u}_i] &= \frac{1}{\|\mathbf{v}\|} \sum_{j=1}^{m} \left( X_{i,j} \frac{\hat{v}_j}{\|\hat{\mathbf{v}}\|} \right) - \frac{1}{\|\mathbf{v}\|} \sum_{j=1}^{m} \left( u_i \mathbb{E}\left[ \frac{\hat{v}_j}{\|\hat{\mathbf{v}}\|} \right] \right) \\
&= \frac{1}{\|\mathbf{v}\|} \sum_{j=1}^{m} \left( X_{i,j} \mathbb{E}\left[ \frac{\hat{v}_j}{\|\hat{\mathbf{v}}\|} \right] + X_{i,j} \left( \frac{\hat{v}_j}{\|\hat{\mathbf{v}}\|} - \mathbb{E}\left[ \frac{\hat{v}_j}{\|\hat{\mathbf{v}}\|} \right] \right) - u_i \mathbb{E}\left[ \frac{\hat{v}_j}{\|\hat{\mathbf{v}}\|} \right] \right) \\
&= \frac{1}{\|\mathbf{v}\|} \sum_{j=1}^{m} (X_{i,j} - u_i) \mathbb{E}\left[ \frac{\hat{v}_j}{\|\hat{\mathbf{v}}\|} \right] + \frac{1}{\|\mathbf{v}\|} \sum_{j=1}^{m} X_{i,j} \left( \frac{\hat{v}_j}{\|\hat{\mathbf{v}}\|} - \mathbb{E}\left[ \frac{\hat{v}_j}{\|\hat{\mathbf{v}}\|} \right] \right).
\end{aligned} \tag{21}
$$

From the triangle inequality, we then have

$$
\begin{aligned}
|\hat{u}_i - \mathbb{E}[\hat{u}_i]| &\leq \frac{1}{\|\mathbf{v}\|} \left| \sum_{j=1}^{m} (X_{i,j} - u_i) \mathbb{E}\left[ \frac{\hat{v}_j}{\|\hat{\mathbf{v}}\|} \right] \right| + \frac{1}{\|\mathbf{v}\|} \sum_{j=1}^{m} |X_{i,j}| \left| \frac{\hat{v}_j}{\|\hat{\mathbf{v}}\|} - \mathbb{E}\left[ \frac{\hat{v}_j}{\|\hat{\mathbf{v}}\|} \right] \right| \\
&\leq \frac{1}{\|\mathbf{v}\|} \left| \sum_{j=1}^{m} (X_{i,j} - u_i) \mathbb{E}\left[ \frac{\hat{v}_j}{\|\hat{\mathbf{v}}\|} \right] \right| + \frac{1}{\|\mathbf{v}\|} \left\| \frac{\hat{\mathbf{v}}}{\|\hat{\mathbf{v}}\|} - \mathbb{E}\left[ \frac{\hat{\mathbf{v}}}{\|\hat{\mathbf{v}}\|} \right] \right\|_1
\end{aligned} \tag{22}
$$

From the fact that $\|\mathbf{v}\| \geq c\sqrt{m}$ and the fact that $\|\mathbf{x}\|_1 \leq \sqrt{m}\|\mathbf{x}\|_2$ for any $\mathbf{x} \in \mathbb{R}^m$, the second term can be bounded as

$$
\begin{aligned}
\frac{1}{\|\mathbf{v}\|} \left\| \frac{\hat{\mathbf{v}}}{\|\hat{\mathbf{v}}\|} - \mathbb{E}\left[ \frac{\hat{\mathbf{v}}}{\|\hat{\mathbf{v}}\|} \right] \right\|_1 &\leq \frac{1}{c} \left\| \frac{\hat{\mathbf{v}}}{\|\hat{\mathbf{v}}\|} - \mathbb{E}\left[ \frac{\hat{\mathbf{v}}}{\|\hat{\mathbf{v}}\|} \right] \right\| \\
&\leq \frac{1}{c} \left\| \frac{\hat{\mathbf{v}}}{\|\hat{\mathbf{v}}\|} - \frac{\mathbf{v}}{\|\mathbf{v}\|} \right\| + \frac{1}{c} \left\| \frac{\mathbf{v}}{\|\mathbf{v}\|} - \mathbb{E}\left[ \frac{\hat{\mathbf{v}}}{\|\hat{\mathbf{v}}\|} \right] \right\| \\
&= \frac{1}{c} \left\| \frac{\hat{\mathbf{v}}}{\|\hat{\mathbf{v}}\|} - \frac{\mathbf{v}}{\|\mathbf{v}\|} \right\| + \frac{1}{c} \left\| \mathbb{E}\left[ \frac{\mathbf{v}}{\|\mathbf{v}\|} - \frac{\hat{\mathbf{v}}}{\|\hat{\mathbf{v}}\|} \right] \right\| \\
&\leq \frac{1}{c} \left\| \frac{\hat{\mathbf{v}}}{\|\hat{\mathbf{v}}\|} - \frac{\mathbf{v}}{\|\mathbf{v}\|} \right\| + \frac{1}{c} \mathbb{E} \left\| \frac{\mathbf{v}}{\|\mathbf{v}\|} - \frac{\hat{\mathbf{v}}}{\|\hat{\mathbf{v}}\|} \right\|,
\end{aligned} \tag{23}
$$

where the last step follows from Jensen's inequality.

Now we consider the second term in (19). We first notice that

$$\hat{u}_i = X_{i,\cdot} \frac{\hat{\mathbf{v}}}{\|\hat{\mathbf{v}}\|\|\mathbf{v}\|} = \frac{X_{i,\cdot}}{\|\mathbf{v}\|}\left(\frac{\mathbf{v}}{\|\mathbf{v}\|} + \frac{\hat{\mathbf{v}}}{\|\hat{\mathbf{v}}\|} - \frac{\mathbf{v}}{\|\mathbf{v}\|}\right) = \frac{X_{i,\cdot}\mathbf{v}}{\|\mathbf{v}\|^2} + \frac{X_{i,\cdot}}{\|\mathbf{v}\|}\left(\frac{\hat{\mathbf{v}}}{\|\hat{\mathbf{v}}\|} - \frac{\mathbf{v}}{\|\mathbf{v}\|}\right),$$

where we recognize the first term as the matched filter for estimating $u_i$ if $\mathbf{v}$ were known. Since $E[X_{i,\cdot}] = u_i\mathbf{v}^T$ and $\hat{\mathbf{v}}$ is independent of $X_{i,\cdot}$ (due to the splitting of the data matrix $X$), we have

$$\mathbb{E}[\hat{u}_i] = \frac{u_i\mathbf{v}^T\mathbf{v}}{\|\mathbf{v}\|^2} + \frac{u_i\mathbf{v}^T}{\|\mathbf{v}\|}\mathbb{E}\left[\frac{\hat{\mathbf{v}}}{\|\hat{\mathbf{v}}\|} - \frac{\mathbf{v}}{\|\mathbf{v}\|}\right] = u_i + \frac{u_i\mathbf{v}^T}{\|\mathbf{v}\|}\left(\mathbb{E}\left[\frac{\hat{\mathbf{v}}}{\|\hat{\mathbf{v}}\|}\right] - \frac{\mathbf{v}}{\|\mathbf{v}\|}\right).$$

Using the Cauchy-Schwarz inequality, we have that

$$|\mathbb{E}[\hat{u}_i] - u_i| \le \frac{u_i\|\mathbf{v}\|}{\|\mathbf{v}\|}\left\|\mathbb{E}\left[\frac{\hat{\mathbf{v}}}{\|\hat{\mathbf{v}}\|}\right] - \frac{\mathbf{v}}{\|\mathbf{v}\|}\right\| = u_i\left\|\mathbb{E}\left[\frac{\hat{\mathbf{v}}}{\|\hat{\mathbf{v}}\|}\right] - \frac{\mathbf{v}}{\|\mathbf{v}\|}\right\| \le u_i\mathbb{E}\left\|\frac{\hat{\mathbf{v}}}{\|\hat{\mathbf{v}}\|} - \frac{\mathbf{v}}{\|\mathbf{v}\|}\right\|, \quad (24)$$

where the last step follows from Jensen's inequality.

Finally, putting together (19), (22), (23), and (24), and noting that $u_i < 1$, we obtain

$$|\hat{u}_i - u_i| \le \frac{1}{\|\mathbf{v}\|}\left|\sum_{j=1}^{m}(X_{i,j} - u_i)E\left[\frac{\hat{v}_j}{\|\hat{\mathbf{v}}\|}\right]\right| + \frac{1}{c}\left\|\frac{\hat{\mathbf{v}}}{\|\hat{\mathbf{v}}\|} - \frac{\mathbf{v}}{\|\mathbf{v}\|}\right\| + \left(1 + \frac{1}{c}\right)\mathbb{E}\left\|\frac{\hat{\mathbf{v}}}{\|\hat{\mathbf{v}}\|} - \frac{\mathbf{v}}{\|\mathbf{v}\|}\right\|.$$

## C  Proof of Equation (11)

We claim that for any $\epsilon > 0$,

$$\mathbb{P}\left(\left|\sum_{j=1}^{m}(X_{i,j} - u_i)\mathbb{E}[\hat{v}_j/\|\hat{\mathbf{v}}\|]\right| > \|\mathbf{v}\|\epsilon\right) \le 2\exp\left(-2c^2m\epsilon^2\right). \quad (25)$$

First we notice that the random variables $(X_{i,j} - u_i)\mathbb{E}\left[\frac{\hat{v}_j}{\|\hat{\mathbf{v}}\|}\right]$, for $j = 1, \dots, m$, are independent and zero-mean. Moreover, they satisfy

$$-u_i\mathbb{E}\left[\frac{\hat{v}_j}{\|\hat{\mathbf{v}}\|}\right] < (X_{i,j} - u_i)\mathbb{E}\left[\frac{\hat{v}_j}{\|\hat{\mathbf{v}}\|}\right] < (1 - u_i)\mathbb{E}\left[\frac{\hat{v}_j}{\|\hat{\mathbf{v}}\|}\right].$$

Using Hoeffding's inequality, for any $\epsilon > 0$ we have that

$$\mathbb{P}\left(\left|\sum_{j=1}^{m}(X_{i,j} - u_i)\mathbb{E}\left[\frac{\hat{v}_j}{\|\hat{\mathbf{v}}\|}\right]\right| > \|\mathbf{v}\|\epsilon\right) \le 2\exp\left(-\frac{2\|\mathbf{v}\|^2\epsilon^2}{\sum_{j=1}^{m}\mathbb{E}[\hat{v}_j/\|\hat{\mathbf{v}}\|]^2}\right)$$

$$\le 2\exp\left(-\frac{2c^2m\epsilon^2}{\mathbb{E}\left[\sum_{j=1}^{m}\hat{v}_j^2/\|\hat{\mathbf{v}}\|^2\right]}\right)$$

$$= 2\exp\left(-2c^2m\epsilon^2\right),$$

where in the second step we used Jensen's inequality.

## D  Proof of Lemma 2

Let $W = X - \mathbb{E}X$ be the "noise" added to $\mathbb{E}X$. In order to prove Lemma 2, our first order of business is to bound the expectation of $\|W\|_{op}$ and $\|W\|_{op}^2$. Then we use these bounds with the Davis-Kahan theorem of Davis and Kahan [15] to bound the $\ell_2$ error in $\hat{\mathbf{v}}$. We have the following lemma.

**Lemma 3.** *The noise matrix $W = X - \mathbb{E}X$ satisfies*

$$\mathbb{E}\left[\|W\|_{op}\right] \le 2\sqrt{(m+n)\log(m+n)} \quad (26)$$

$$\mathbb{E}\left[\|W\|_{op}^2\right] \le 120(m+n)\log(m+n) \quad (27)$$

*Proof.* Strictly speaking, due to Jensen's inequality, (27) implies (26) (with a different constant). However, to provide intuition and improve the exposition, we provide a standalone proof of (26) first. We start by noticing that

$$W_{i,j} = \begin{cases} 1 - u_i v_j & \text{with probability } u_i v_j, \\ -u_i v_j & \text{with probability } 1 - u_i v_j, \end{cases}$$

Hence we have

$$(\mathbb{E}[W^T W])_{i,j} = \begin{cases} 0 & \text{if } i \neq j, \\ \sum_{k=1}^n u_k v_i (1 - u_k v_i) & \text{if } i = j, \end{cases}$$

which implies that $c^2(1 - C^2)n \leq (\mathbb{E}[W^T W])_{i,i} \leq C^2(1 - c^2)n$. Thus $\|\mathbb{E}[W^T W]\|_{op} \in [c^2(1 - C^2)n, C^2(1 - c^2)n]$. Similarly one can argue that $\|\mathbb{E}[WW^T]\|_{op} \in [c^2(1 - C^2)m, C^2(1 - c^2)m]$. Following the notation of Tropp [43, Theorem 6.1.1], the matrix variance statistic $\nu(W)$ is

$$\begin{aligned} \nu(W) &= \max\left(\|\mathbb{E}[W^T W]\|_{op}, \|\mathbb{E}[WW^T]\|_{op}\right), \\ &\in [c^2(1 - C^2)\max(m, n), C^2(1 - c^2)\max(m, n)], \\ &\leq C^2(1 - c^2)(m + n) \leq m + n. \end{aligned} \tag{28}$$

From the Matrix-Bernstein inequality [43, Eq. (6.1.3)], we have that

$$\begin{aligned} \mathbb{E}\|W\|_{op} &\leq \sqrt{2\nu(W)\log(m + n)} + \tfrac{1}{3}\log(m + n) \\ &= \sqrt{2(m + n)\log(m + n)} + \tfrac{1}{3}\log(m + n) \\ &\leq (\sqrt{2} + \tfrac{1}{3})\sqrt{(m + n)\log(m + n)} \leq 2\sqrt{(m + n)\log(m + n)}, \end{aligned}$$

proving (26). To prove (27), we rely on another inequality by Tropp [43, Eq. (6.1.6)] to state that

$$\begin{aligned} \left(\mathbb{E}\|W\|_{op}^2\right)^{1/2} &\leq \sqrt{2e\nu(W)\log(m + n)} + 4e\log(m + n) \\ &\leq \sqrt{2e(m + n)\log(m + n)} + 4e\log(m + n) \\ &\leq 4e\sqrt{(m + n)\log(m + n)}, \end{aligned}$$

which implies that $\mathbb{E}\|W\|_{op}^2 \leq 120(m + n)\log(m + n)$, proving (27). $\qquad\square$

With Lemma 3, we proceed to the proof of Lemma 2. Notice that the leading right singular vector of $X$ is equivalent to the leading eigenvector of $X^T X$. Also note that

$$\begin{aligned} X^T X &= (\mathbb{E}X + W)^T(\mathbb{E}X + W) \\ &= (\mathbf{u}\mathbf{v}^T + W)^T(\mathbf{u}\mathbf{v}^T + W) \\ &= \|\mathbf{u}\|^2 \mathbf{v}\mathbf{v}^T + \mathbf{v}\mathbf{u}^T W + W^T \mathbf{u}\mathbf{v}^T + W^T W. \end{aligned} \tag{29}$$

We will use the Davis-Kahan theorem to bound $\left\|\frac{\hat{\mathbf{v}}}{\|\hat{\mathbf{v}}\|} - \frac{\mathbf{v}}{\|\mathbf{v}\|}\right\|$. We begin by bounding the operator norm of the "error terms" in (29) as

$$\begin{aligned} \|\mathbf{v}\mathbf{u}^T W + W^T \mathbf{u}\mathbf{v}^T + W^T W\|_{op} &\overset{(a)}{\leq} \|\mathbf{v}\mathbf{u}^T W\|_{op} + \|W\mathbf{u}\mathbf{v}^T\|_{op} + \|W^T W\|_{op}, \\ &\overset{(b)}{\leq} \|\mathbf{v}\mathbf{u}^T\|_{op}\|W\|_{op} + \|W\|_{op}\|\mathbf{u}\mathbf{v}^T\|_{op} + \|W^T W\|_{op}, \\ &= 2\|\mathbf{u}\|\|\mathbf{v}\|\|W\|_{op} + \|W^T W\|_{op}, \end{aligned} \tag{30}$$

where $(a)$ follows from the triangle inequality, and $(b)$ from the fact that $\|AB\|_{op} \leq \|A\|_{op}\|B\|_{op}$. We also note that

$$c\sqrt{n} \leq \|\mathbf{u}\| \leq C\sqrt{n} \leq \sqrt{n}, \tag{31}$$
$$c\sqrt{m} \leq \|\mathbf{v}\| \leq C\sqrt{m} \leq \sqrt{m}. \tag{32}$$

Since $\|\mathbf{u}\|^2\mathbf{v}\mathbf{v}^T$ is rank-one with leading eigenvalue $\|\mathbf{u}\|^2\|\mathbf{v}\|^2$, the spectral gap $\delta$ of $\|\mathbf{u}\|^2\mathbf{v}\mathbf{v}^T$ is $\delta = \|\mathbf{u}\|^2\|\mathbf{v}\|^2 \geq c^4(mn)$. From the version of the Davis-Kahan Theorem [15] of Mahoney [33, Theorem 30], we have that

$$
\left\| \frac{\hat{\mathbf{v}}}{\|\hat{\mathbf{v}}\|} - \frac{\mathbf{v}}{\|\mathbf{v}\|} \right\| \leq \sqrt{2}\frac{\|\mathbf{v}\mathbf{u}^T W + W^T\mathbf{u}\mathbf{v}^T + W^T W\|_{op}}{\delta}
$$

$$
\leq \frac{\sqrt{8mn}\|W\|_{op} + \sqrt{2}\|W^T W\|_{op}}{c^4(mn)} = \frac{\sqrt{8}\|W\|_{op}}{c^4\sqrt{mn}} + \frac{\sqrt{2}\|W\|_{op}^2}{c^4 mn}. \tag{33}
$$

Taking the expectation on both sides and using Lemma 3, we obtain

$$
\mathbb{E}\left\| \frac{\hat{\mathbf{v}}}{\|\hat{\mathbf{v}}\|} - \frac{\mathbf{v}}{\|\mathbf{v}\|} \right\| \leq \frac{\sqrt{16(m+n)\log(m+n)}}{c^4\sqrt{mn}} + \frac{120\sqrt{2}(m+n)\log(m+n)}{c^4 mn}
$$

$$
= \frac{4}{c^4}\sqrt{\left(\frac{1}{m}+\frac{1}{n}\right)\log(m+n)} + \frac{120\sqrt{2}}{c^4}\left(\frac{1}{m}+\frac{1}{n}\right)\log(m+n)
$$

$$
\leq \frac{4}{c^4}\sqrt{\frac{\log(m+n)}{\min(m,n)}} + \frac{120\sqrt{2}}{c^4}\frac{\log(m+n)}{\min(m,n)} \tag{34}
$$

Notice that $\hat{\mathbf{v}}/\|\hat{\mathbf{v}}\|$ and $\mathbf{v}/\|\mathbf{v}\|$ are unit vectors, and so their $\ell_2$ distance can be at most 2. The right-hand side of (34) can only be less than 2 if

$$
\frac{4}{c^4}\sqrt{\frac{\log(m+n)}{\min(m,n)}} \leq 1 \;\Rightarrow\; \frac{\log(m+n)}{\min(m,n)} \leq \frac{c^4}{4}\sqrt{\frac{\log(m+n)}{\min(m,n)}}.
$$

Hence,

$$
\mathbb{E}\left\| \frac{\hat{\mathbf{v}}}{\|\hat{\mathbf{v}}\|} - \frac{\mathbf{v}}{\|\mathbf{v}\|} \right\| \leq \min\left[2, \frac{4}{c^4}\sqrt{\frac{\log(m+n)}{\min(m,n)}} + \frac{120\sqrt{2}}{c^4}\frac{\log(m+n)}{\min(m,n)}\right]
$$

$$
\leq \left(\frac{4}{c^4} + 30\sqrt{2}\right)\sqrt{\frac{\log(m+n)}{\min(m,n)}}. \tag{35}
$$

This proves Statement (b) in Lemma 2, where we can take $C_3 = 4/c^4 + 30\sqrt{2}$. To prove Statement (a), we note that from (33),

$$
\left\| \frac{\hat{\mathbf{v}}}{\|\hat{\mathbf{v}}\|} - \frac{\mathbf{v}}{\|\mathbf{v}\|} \right\| \leq \frac{\sqrt{8}\|W\|_{op}}{\|\mathbf{u}\|\|\mathbf{v}\|} + \frac{\sqrt{2}\|W\|_{op}^2}{\|\mathbf{u}\|^2\|\mathbf{v}\|^2} \tag{36}
$$

Next we notice that, if $\frac{\|W\|_{op}}{\|\mathbf{u}\|\|\mathbf{v}\|} < \epsilon/4$, then

$$
\left\| \frac{\hat{\mathbf{v}}}{\|\hat{\mathbf{v}}\|} - \frac{\mathbf{v}}{\|\mathbf{v}\|} \right\| < \frac{\epsilon}{\sqrt{2}} + \frac{\epsilon^2}{8\sqrt{2}} < \epsilon,
$$

for $0 < \epsilon < 1$. Therefore, we have that

$$
\mathbb{P}\left(\left\| \frac{\hat{\mathbf{v}}}{\|\hat{\mathbf{v}}\|} - \frac{\mathbf{v}}{\|\mathbf{v}\|} \right\| \geq \epsilon\right) \leq \mathbb{P}\left(\|W\|_{op} > \tfrac{1}{4}\epsilon\|\mathbf{u}\|\|\mathbf{v}\|\right)
$$

$$
\leq \mathbb{P}\left(\|W\|_{op} > \tfrac{1}{4}\epsilon c^2\sqrt{mn}\right)
$$

$$
\leq (m+n)\exp\left(-\frac{1}{32}\frac{c^4\epsilon^2 mn}{m+n+\frac{1}{12}c^2\epsilon\sqrt{mn}}\right) \tag{37}
$$

$$
\leq (m+n)\exp\left(-\frac{1}{32}\frac{c^4\epsilon^2 mn}{(2+\frac{1}{12})\max(m,n)}\right) \tag{38}
$$

$$
\leq (m+n)\exp\left(-\frac{c^4\epsilon^2}{48}\min(m,n)\right)
$$

where (37) follows by the Matrix-Bernstein inequality [43, Eq. (6.1.4)] using the computation of the matrix variance statistic from (28), and (38) follows since $c < 1$ and $\epsilon < 1$. This means we can take $C_2 = c^4/48$.

# E  Thresholding Bandits

In this section, we develop a bandit algorithm to return a set of coordinates in $\{1, \ldots, n\}$ such that with high probability all coordinates with $u_i \geq \beta$ are returned and no coordinates with $u_i \leq \alpha$ are returned, for some $\beta > \alpha$. We assume that $\beta - \alpha > \sqrt{\frac{12 \log n}{C_4 n}}$. The algorithm and analysis follows Locatelli et al. [32].

**Lemma 4.** *For any $\Gamma \in \mathbb{R}$, our estimates $\hat{u}_i$ from Algorithm 1 run with an $n \times m$ matrix with $m, n$ such that*

$$\min(m, n) \geq \frac{\log\left(\frac{1}{\delta}\right) + \log(m + n + 2)}{\Gamma^2 C_4}, \tag{39}$$

*will have $|\hat{u}_i - u_i| \leq \Gamma$ with probability at least $1 - \delta$. Thus, if*

$$\min(m, n) \geq \frac{3 \log n + 2 \log(m + n)}{\Gamma^2 C_4} \tag{40}$$

*then with probability $1 - \frac{1}{n^2}$, $|\hat{u}_i - u_i| \leq \Gamma$ for all $i \in \{1, \ldots, n\}$.*

*Proof.* Eq. (39) follows by inverting the error bound from Eq. (12). Eq. (40) follows by just substituting $\delta = \frac{1}{n^3}$ and noting that $\log(m + n + 2) \leq 2 \log(m + n)$ as long as $m + n \geq 4$. $\qquad\square$

Notice that a naive non-adaptive approach to this thresholding problem would consist of applying Algorithm 1 to the entire observation matrix, with enough enough workers such that the confidence intervals are smaller than $\frac{\beta - \alpha}{2}$ and return coordinates with value more than $\frac{\alpha + \beta}{2}$. In that case, Lemma 4 implies that $\frac{12}{C_4} \frac{n \log n}{(\beta - \alpha)^2}$ workers' responses are enough to succeed with probability at least $1 - \frac{1}{n}$. In Algorithm 3, we propose an adaptive way of performing the same task.

---

**Algorithm 3** Adaptive Spectral Thresholding Algorithm

---

1: **Input:** Range $[\alpha, \beta]$.
2: Initialise $S_0 \leftarrow \{1, \ldots, n\}$          $\triangleright$ Set of coordinates initially under consideration
3: Initialise $A \leftarrow \emptyset$          $\triangleright$ Set of coordinates initially accepted
4: $t_{-1} \leftarrow \frac{12 \log n}{C_4}$          $\triangleright$ Initial number of workers recruited. $C_4$ from Eq. (39)
5: **for** $r = 0$ **to** $\lceil \log_2 \frac{1}{\beta - \alpha} \rceil - 1$ **do**
6:      $t_r = 4 t_{r-1}$          $\triangleright$ Number of workers to be recruited
7:      Obtain a binary response matrix $X^{(r)} \in \{0, 1\}^{|S_r| \times t_r}$ and corresponding $\|\mathbf{v}^{(r)}\|$
8:      Use Algorithm 1 to compute estimates $\hat{\mathbf{u}}^{(r)}$ for $X^{(r)}$
9:      Construct confidence intervals $C(t_r)$ (same for every question)
10:     Construct accepted set $C_{\text{acc}} = \{i \in S_r : \hat{u}_i^{(r)} - C(t_r) > \alpha\}$
11:     Construct rejected set $C_{\text{rej}} = \{i \in S_r : \hat{u}_i^{(r)} + C(t_r) < \beta\}$
12:     Set $S_{r+1} = S_r \setminus \{C_{\text{rej}} \cup C_{\text{acc}}\}$
13:     **if** $|S_{r+1}| < \frac{12}{(\beta - \alpha)^2 C_4} \log n$ **then**
14:        Let $\mathcal{I}$ be $\frac{12}{(\beta - \alpha)^2 C_4} \log n - |S_{r+1}|$ coordinates of $C_{\text{rej}} \cup C_{\text{acc}}$ picked uniformly at random.
15:        $S_C = S_{r+1} \cup \mathcal{I}$
16:        $A = A \cup (C_{\text{acc}})$
17:        **break**
18:     **else**
19:        Set $A = A \cup C_{\text{acc}}$
20: **Clean up:** Use $t_C = \frac{12 \log n}{(\beta - \alpha)^2 C_4}$ workers with set $S_C$ of questions to obtain estimates $\hat{\mathbf{u}}^{(C)}$.

     Construct confidence intervals $C(t_C)$ and accepted set $C_A = \{i \in S_C : \hat{u}_i^{(C)} - C(t_C) > \alpha\}$
21: **return** $A \cup C_A$.

---

Let $\kappa \triangleq \left\lfloor \frac{12}{(\beta-\alpha)^2 C_4} \log n \right\rfloor$. Define

$$\Gamma_i = \begin{cases} u_i - \alpha & \text{if } \beta < u_i, \\ \beta - \alpha & \text{if } \alpha \leq u_i \leq \beta, \\ \beta - u_i & \text{if } u_i < \alpha. \end{cases} \tag{41}$$

See Figure 3 for an illustration. Further let $\Gamma^{(1)} \leq \Gamma^{(2)} \leq \cdots \leq \Gamma^{(n)}$ be the sorted list of the $\Gamma_i$.

Figure 3: An illustration of $\Gamma_i$ of Eq. (41)

**Theorem 4** (Restatement of Theorem 3). *Given parameters $\beta$ and $\alpha$ such that $\beta - \alpha > \sqrt{\frac{12 \log n}{C_4 n}}$, Algorithm 3 will output a set of reads $R$ such that $\{i : u_i > \beta\} \subseteq R \subseteq \{i : u_i > \alpha\}$ with probability at least $1 - \frac{2}{n}$, requiring a budget of at most*

$$T = 2 \left( \frac{12}{C_4} \frac{\log n}{(\beta - \alpha)^2} \right)^2 + \sum_{\ell=\kappa+1}^{n} \frac{32}{C_4} \frac{\log n}{(\Gamma^{(\ell)})^2} \tag{42}$$

*on this success event.*

*Proof.* We note that while in the elimination stage, we always have $t_r \leq |S_r|$ by construction, as $t_0 \leq |S_0|$ and $t_r$ is an increasing sequence while $|S_r|$ is a decreasing one, and we break when $|S_r| < \frac{12}{(\beta-\alpha)^2 C_4} \log n$, while the maximum $t_r$ achieved in the **for** loop of line $r$ is $\frac{12}{(\beta-\alpha)^2 C_4} \log n$ because of the limits of the for loop. Further $t_r$ is picked so that in round $r$, with probability $1 - \frac{1}{n^2}$, $|\hat{u}_i^{(r)} - u_i| \leq 2^{-r}$ for all $i \in \{1, \ldots, n\}$. Notice that there are at most $\lceil \log_2 \frac{1}{\beta-\alpha} \rceil$ iterations and, since $\beta - \alpha$ is assumed to be a constant, $\lceil \log_2 \frac{1}{\beta-\alpha} \rceil \leq n$ for large enough $n$. Hence, with probability at least $1 - \frac{1}{n}$, we have that over all rounds, $\hat{u}_i^{(r)}$ are within their confidence intervals for all coordinates $i$. Similarly the clean up stage is constructed so that $|\hat{u}_i^{(r)} - u_i| \leq \beta - \alpha$ with probability at least $1 - \frac{1}{n^2}$. Thus with probability at least $1 - \frac{2}{n}$ all our estimates are within the confidence intervals constructed throughout the algorithm.

Notice that if the confidence interval of question $i$ (with parameter $u_i$) is reduced to less than $\Gamma_i/2$ in the $r$-th iteration, at that point (or previously) the algorithm must either accept or reject $u_i$. This is because any $u_i > \beta$ will have $\hat{u}_i^{(r)} - \Gamma_i/2 > \alpha$, any $u_i < \alpha$ will have $\hat{u}_i^{(r)} + \Gamma_i/2 < \beta$, and any $u_i \in [\alpha, \beta]$ will have a confidence interval of total length at most $\Gamma_i = \beta - \alpha$, which cannot include both $\alpha$ and $\beta$. Hence, for each question $i$ of the $n - \kappa$ questions eliminated before the clean up stage, the total number of workers used at its last iteration before elimination is, by Lemma 4, at most

$$\frac{3 \log n + 2 \log(2n)}{C_4(\Gamma_i/2)^2} \leq \frac{24 \log n}{C_4 \Gamma_i^2},$$

where we upper bound $2\log(2n)$ by $3\log(n)$ for $n > 4$. Since the number of workers used at each iteration grows as a geometric progression, the total number of questions answered for all the $n - \kappa$ questions that are eliminated before the clean up stage is at most

$$\sum_{\ell=\kappa+1}^{n} \frac{32}{C_4} \frac{\log n}{(\Gamma^{(\ell)})^2}.$$

Similarly for each question resolved in the clean up stage of $\kappa$ workers, we need at most $\frac{24}{C_4} \frac{\log n}{(\beta-\alpha)^2}$ total worker responses. Since

$$\kappa \frac{24}{C_4} \frac{\log n}{(\beta - \alpha)^2} \leq 2 \left( \frac{12}{C_4} \frac{\log n}{(\beta - \alpha)^2} \right)^2,$$

the result follows. $\qquad\square$

**Remark 1.** *Line 3 is simply to make sure that we have $n > m$ in the clean up stage. Selecting $\mathcal{I}$ uniformly at random from $C_{\text{rej}} \cup C_{\text{acc}}$ is simply given as a concrete way for the algorithm to run.*

**Remark 2.** *Theorem 4 basically enables us to construct confidence intervals of $\frac{\beta-\alpha}{2}$ for only the "hardest" $\kappa$ questions and near optimal confidence intervals for the rest, while the non-adaptive algorithm would need to construct confidence intervals of $\frac{\beta-\alpha}{2}$ for all.*

## F   Proof of Theorem 2

The proof at a high level proceeds by showing that the probability we eliminate any of the top $k$ arms is low in our halving stages, and then that uniformly sampling the $\sqrt{T}$ remaining arms allows us to identify the top $k$ correctly. Note that coordinates, items, and arms will be used interchangeably. For the sake of notational simplicity, we assume that the arms are sorted by mean, in that $\mu_1 \geq \ldots \geq \mu_n$, and that $\mu_k > \mu_{k+1}$, i.e. the top-$k$ are well defined. We begin by observing that we do not exceed our allotted budget.

**Lemma 5.** *Algorithm 2 does not exceed the budget $T$.*

*Proof.* At the $r$-th stage we have $|\mathcal{I}_r|$ questions and $t_r$ workers. Hence,

$$\sum_{r=0}^{r_{\max}} |\mathcal{I}_r| t_r \leq \sum_{r=0}^{r_{\max}} \frac{T}{2\left\lceil \log_2 \frac{n}{\sqrt{T}} \right\rceil} = \frac{T(r_{\max}+1)}{2\left\lceil \log_2 \frac{n}{\sqrt{T}} \right\rceil} = \frac{T\left\lceil \log_2 \frac{n}{\sqrt{T}} \right\rceil}{2\left\lceil \log_2 \frac{n}{\sqrt{T}} \right\rceil} = \frac{T}{2}$$

Since the clean up stage uses at most $\frac{T}{2}$ pulls, the algorithm does not exceed its budget of $T$ pulls. $\quad\square$

We now examine one round of our adaptive spectral algorithm and bound the probability that the algorithm eliminates one of the top-$k$ arms in round $r$, recalling that

$$t_r = \left\lfloor \frac{T}{2|\mathcal{I}_r|\lceil \log_2 \frac{n}{\sqrt{T}} \rceil} \right\rfloor.$$

In standard bandit analyses, we obtain concentration of our estimated arm means via Hoeffding's inequality, which we are unable to utilize here. Theorem 1 states that

$$\mathbb{P}(|\hat{u}_i - u_i| \geq \epsilon) \leq 3n \exp\left(-C_1 \epsilon^2 m\right),$$

providing a Hoeffding-like bound that allows us to eliminate suboptimal arms with good probability.

**Lemma 6.** *The probability that one of the top $k$ arms is eliminated in round $r$ is at most*

$$18kn \exp\left(-C_5 \frac{\Delta_{i_r}^2}{i_r} \frac{T}{\log \frac{n}{\sqrt{T}}}\right)$$

*for $i_r = |\mathcal{I}_r|/4 = \frac{n}{2^{r+2}}$, and $C_5 = \frac{C_1}{64}$.*

*Proof.* The proof follows similarly to that of [28]. To begin, define $\mathcal{I}'_r$ as the set of coordinates in $\mathcal{I}_r$ excluding the $i_r = \frac{1}{4}|\mathcal{I}_r|$ coordinates $i$ with largest $u_i$. Let $\hat{u}_i^{(r)}$ be the estimator of $u_i$ in round $r$. We define the random variable $N_r$ as the number of arms in $\mathcal{I}'_r$ whose $\hat{u}_i^{(r)}$ in round $r$ is larger than that of any of the top-$k$ $u_i$. We begin by showing that $\mathbb{E}[N_r]$ is small. We bound $\mathbb{E}[N_r]$ as

$$
\mathbb{E}[N_r] = \sum_{i \in \mathcal{I}'_r} \mathbb{P}\left(\bigcup_{\ell \in [k]} \left\{\hat{u}_i^{(r)} \geq \hat{u}_\ell^{(r)}\right\}\right) \leq k \sum_{i \in \mathcal{I}'_r} \mathbb{P}\left(\hat{u}_i^{(r)} \geq \hat{u}_k^{(r)}\right)
$$

$$
\leq k \sum_{i \in \mathcal{I}'_r} \mathbb{P}\left(\hat{u}_i^{(r)} \geq u_i + \Delta_i/2\right) + \mathbb{P}\left(\hat{u}_k^{(r)} < u_k - \Delta_i/2\right)
$$

$$
\leq k|\mathcal{I}'_r| \left(\mathbb{P}\left(\hat{u}_{i_r}^{(r)} \geq u_{i_r} + \Delta_{i_r}/2\right) + \mathbb{P}\left(\hat{u}_k^{(r)} < u_k - \Delta_{i_r}/2\right)\right)
$$

$$
\leq 6k|\mathcal{I}'_r||\mathcal{I}_r| \exp\left(-\frac{C_1}{4}\Delta_{i_r}^2 t_r\right)
$$

$$
\leq 6k|\mathcal{I}'_r|n \exp\left(-\frac{C_1}{64}\frac{\Delta_{i_r}^2}{i_r}\frac{T}{\log \frac{n}{\sqrt{T}}}\right),
$$

We note that since the $i_r$ largest entries of $\mathcal{I}$ are not present in $\mathcal{I}'_r$, we have that $\max_{i \in \mathcal{I}'_r} u_i \leq u_{i_r}$. We now see that in order for one of the top $k$ arms to be eliminated in round $r$, at least $|\mathcal{I}_r|/2$ arms must have had higher empirical scores in round $r$ than it. This means that at least $|\mathcal{I}_r|/4$ arms from $\mathcal{I}'_r$ must outperform the top $k$ arms, i.e., $N_r \geq |\mathcal{I}_r|/4 = |\mathcal{I}'_r|/3$. Note that this analysis only holds when $|\mathcal{I}_r| \geq 4k$. We can then bound this probability with Markov's inequality as

$$
\mathbb{P}\left(\begin{array}{c}\text{At least one of top-}k\text{ arms} \\ \text{eliminated in round }r\end{array}\middle|\begin{array}{c}\text{None of the top }k\text{ arms} \\ \text{eliminated till round }r\end{array}\right) \leq \mathbb{P}\left(N_r \geq \frac{1}{3}|\mathcal{I}'_r|\right) \leq 3\mathbb{E}[N_r]/|\mathcal{I}'_r|
$$

$$
\leq 18kn \exp\left(-\frac{C_1}{64}\frac{\Delta_{i_r}^2}{i_r}\frac{T}{\log \frac{n}{\sqrt{T}}}\right),
$$

concluding the proof of the lemma. $\qquad\square$

**Lemma 7.** *The total probability of failure during the elimination stages, $P_e$, is bounded as*

$$
P_e \leq 18kn \log \frac{n}{\sqrt{T}} \exp\left(-C_5 \frac{\Delta_{i_r}^2}{i_r}\frac{T}{\log \frac{n}{\sqrt{T}}}\right)
$$

*Proof.* We see by a union bound over the stages that the probability that the algorithm fails (it eliminates one of the top $k$ arms) in any of the $\log \frac{n}{\sqrt{T}}$ halving stages is at most

$$
P_{e1} = \sum_{r=0}^{\log \frac{n}{\sqrt{T}}-1} \mathbb{P}\left(\begin{array}{c}\text{At least one of top-}k\text{ arms} \\ \text{eliminated in round }r\end{array}\middle|\begin{array}{c}\text{None of the top }k\text{ arms} \\ \text{eliminated till round }r\end{array}\right) \mathbb{P}\left(\begin{array}{c}\text{None of the top }k\text{ arms} \\ \text{eliminated till round }r\end{array}\right)
$$

$$
\leq \sum_{r=0}^{\log \frac{n}{\sqrt{T}}-1} \mathbb{P}\left(\begin{array}{c}\text{At least one of top-}k\text{ arms} \\ \text{eliminated in round }r\end{array}\middle|\begin{array}{c}\text{None of the top }k\text{ arms} \\ \text{eliminated till round }r\end{array}\right)
$$

$$
\leq \sum_{r=0}^{\log \frac{n}{\sqrt{T}}-1} 18kn \exp\left(-C_5 \frac{\Delta_{i_r}^2}{i_r}\frac{T}{\log \frac{n}{\sqrt{T}}}\right)
$$

$$
\leq \sum_{r=0}^{\log \frac{n}{\sqrt{T}}-1} 18kn \exp\left(-C_5 \frac{T}{\log \frac{n}{\sqrt{T}} \cdot \max_s \frac{i_s}{\Delta_{i_s}^2}}\right)
$$

$$
\leq 18kn \log \frac{n}{\sqrt{T}} \exp\left(-C_5 \frac{T}{\log \frac{n}{\sqrt{T}} \cdot \max_{i \geq \sqrt{T}} \frac{i}{\Delta_i^2}}\right)
$$

$$
\leq 18kn \log \frac{n}{\sqrt{T}} \exp\left(-C_5 \frac{T}{H_2 \log \frac{n}{\sqrt{T}}}\right),
$$

as $H_2 \triangleq \max_i \frac{i}{\Delta_i^2}$. This concludes the proof of the lemma. $\qquad\square$

**Lemma 8.** *The total probability of failure during the clean up stage, $P_f$, is upper bounded as*

$$P_f \leq 12T \exp\left(-\frac{C_1}{16}\Delta_{k+1}^2 \sqrt{T}\right)$$

*Proof.* Through our halving stages, we are left with at most $2\sqrt{T}$ active coordinates. We now use a budget of $T/2$, i.e. $m \geq \sqrt{T}/4$ columns to estimate their means. Then, the probability that the top $k$ are not the true top $k$ entries (given that none of the top $k$ were eliminated previously) is:

$$
\begin{aligned}
P_f &\leq \sum_{i \in \mathcal{I}_{r_{max}+1}} \mathbb{P}\left(|\hat{u}_i - u_i)| > \Delta_{k+1}/2\right) \\
&\leq 2\sqrt{T} \cdot \mathbb{P}\left(|\hat{u}_1 - u_1)| > \Delta_{k+1}/2\right) \\
&\leq 12T \exp\left(-\frac{C_1}{16}\Delta_{k+1}^2 \sqrt{T}\right)
\end{aligned}
$$

$\qquad\square$

Thus, our overall error probability is at most

$$\mathbb{P}(\text{failure}) \leq P_e + P_f$$

$$
\leq 18kn \log \frac{n}{\sqrt{T}} \exp\left(-C_5 \frac{T}{H_2 \log \frac{n}{\sqrt{T}}}\right) + 12T \exp\left(-\frac{C_1}{16}\Delta_{k+1}^2 \sqrt{T}\right)
$$

$$
\leq 18kn \log n \exp\left(-C_5 \frac{T}{H_2 \log n}\right) + 12n^2 \exp\left(-\frac{C_1}{16}\Delta_{k+1}^2 \sqrt{T}\right)
$$

with budget no more than T.

Inverting this, we have that for a probability of error $\delta$, one needs $T = O\left(H_2 \log n \log\left(\frac{kn \log n}{\delta}\right) + \Delta_{k+1}^{-4} \log^2\left(\frac{n^2}{\delta}\right)\right)$, giving us the result claimed.

## G   Constants

The results in Section 3 are stated in terms of several constants. We assume that there is some $c > 0$ such that $p_i > c$ , $q_j > c$ for all $i, j$. Following the derivations in their respective proofs, these constants are given by

$$
\begin{aligned}
C_1 &= \min(C_4, (6C_3/c)^{-2}) \\
C_2 &= c^4/48 \\
C_3 &= 4/c^4 + 30\sqrt{2} \\
C_4 &= c^2 \min(1/18, C_2/9) \\
C_5 &= \frac{C_1}{64}.
\end{aligned}
$$

We present these constants here for the sake of completeness, noting that several of the bounding steps in the derivations could be loose, and these constants are not expected to be tight. One way to improve algorithm performance in practice is to first run the algorithm in Section 3 on a dataset with known ground truth, and empirically estimate the true constants.

## H   Comparison with other methods

In this section, we discuss an alternative way to construct estimators to be used with the bandit algorithms. We consider row averages as an estimator for the bandit algorithms and show that we could run a top-$k$ algorithm with row averages as the estimators, but would not be able to run the thresholding bandits algorithm as we do not obtain unbiased estimates.

We can construct estimators based off of row sums. For $X$ with $\mathbb{E}X = \mathbf{u}\mathbf{v}^T$, we estimate $u_i$ with

$$\hat{u}_i^{(m)} = \frac{1}{m}\sum_{j=1}^{m} X_{i,j}. \tag{43}$$

Notice that this is similar in spirit to the Jaccard similarity estimator described in (3) and, in practice, provides a worse estimator to the overlap sizes than the estimators based on a rank-one model [5]. However, these estimators can theoretically still be used to find the reads with the largest overlaps, as we describe next. Considering that $X_{i,j}$ has expectation $u_i v_j$, we note that

$$
\begin{aligned}
\hat{u}_i^{(m)} - \hat{u}_k^{(m)} &= \frac{1}{m}\sum_{j=1}^{m}\left(X_{i,j} - X_{k,j}\right) \\
&= \frac{1}{m}\sum_{j=1}^{m}\left((X_{i,j} - u_i v_j) + u_i v_j - (X_{k,j} - u_k v_j) - u_k v_j\right) \\
&= \frac{1}{m}\sum_{j=1}^{m}\left(X_{i,j} - u_i v_j - (X_{k,j} - u_k v_j)\right) + \frac{1}{m}\sum_{j=1}^{m} v_j (u_i - u_k)
\end{aligned}
$$

Hence, for $u_i > u_k$,

$$
\begin{aligned}
\mathbb{P}\left(\hat{u}_i^{(m)} < \hat{u}_k^{(m)}\right) &\leq \mathbb{P}\left(\frac{1}{m}\sum_{j=1}^{m}\left(X_{i,j} - u_i v_j - (X_{k,j} - u_k v_j)\right) < \frac{1}{m}\sum_{j=1}^{m} v_j (u_k - u_i)\right) \\
&\leq \mathbb{P}\left(\frac{1}{2m}\sum_{j=1}^{m}\left(X_{i,j} - u_i v_j - (X_{k,j} - u_k v_j)\right) < \left(\frac{1}{m}\sum_{j=1}^{m} v_j\right)\frac{u_k - u_i}{2}\right) \\
&\leq 2\exp\left(-\frac{m}{4}\left(\frac{1}{m}\sum_{j=1}^{m} v_j\right)^2 (u_k - u_i)^2\right), \\
&= 2\exp\left(-\frac{m\bar{v}^2(u_k - u_i)^2}{4}\right),
\end{aligned}
$$

where $\bar{v} = \frac{1}{m}\sum_{j=1}^{m} v_j$.

This follows since $X_{i,j} - u_i v_j$ is a zero-mean bounded random variable. This bound implies that $m = \frac{4\log\left(\frac{2n}{\delta}\right)}{\bar{v}^2 \epsilon^2}$ yields the desired result, that $\mathbb{P}\left(\operatorname{argmax}_{i\in[n]} \hat{u}_i^{(m)} = \operatorname{argmax}_{i\in[n]} u_i\right) \geq 1 - \delta$. and so for the top-$k$ scenario we have that a budget of

$$T = n\frac{\log\left(\frac{2n}{\delta}\right)}{\bar{v}^2 \Delta_{k+1}^2} \tag{44}$$

is required by the uniform sampling row sum algorithm. We note that this concentration analysis is for uniform sampling, but it shows us that we could do sequential halving on the row sums to adaptively find the maximal $u_i$, with a similar analysis to [4]. Note that this is critically using the fact that the estimators for $u_i$ and $u_k$ are taken across the same $v_j$, and that we are not able to generate unbiased estimates of the $u_i$ with this method, only to preserve ordering. Hence while such an estimator can be used with a top-$k$ bandits algorithm, we are unable to use it with a thresholding bandit algorithm.

**Remark 3.** *To provide some intuition, we remark that the row averages estimator has an advantage, in that for an $n \times m$ matrix $X$ the error of $\hat{u}_i^{(m)} - \hat{u}_k^{(m)}$ decays roughly as $\frac{1}{\sqrt{m}}$ (when $\bar{v}$ is $O(1)$). On the other hand, with the spectral estimator we use, the error of the estimator of $u_i$ only decays as roughly $\frac{1}{\sqrt{\min(n,m)}}$.*

# I   Implementation Details for Algorithm 2

While in theory we stop at $\sqrt{T}$ arms remaining, in practice we continue halving until there are fewer than $2k$ remaining arms, at which point we output the $k$ arms with largest $\hat{u}_i$. While theoretically we

are unable to take advantage of the scenario $m > n$ (due to the constraint in Theorem 1), in practice, increasing $m$ beyond $n$ still improves the estimates $\hat{u}_i$, and we do not need to perform the clean up stage with $\sqrt{T}$ arms remaining.

In the practical implementation of Algorithm 2, we also impose a maximum number of measurements per item (finite number of workers), and so terminate our algorithm and return the top $k$ if $t_r = m_{\max}$ for some a priori fixed quantity $m_{\max}$.

While Algorithm 2 requires "oracle knowledge" of $\|\mathbf{v}^{(r)}\|$, in practice that cannot be obtained and we use $\|\hat{\mathbf{v}}^{(r)}\|$ instead. Notice that knowledge of the exact value of $\|\mathbf{v}\|$ would only provide a rescaling of our estimates, and so relative ordering is preserved in our $\hat{\mathbf{u}}^{(r)}$ if we use $\|\hat{\mathbf{v}}^{(r)}\|$, which is sufficient for top-$k$ identification. Notice that this is not the case for the thresholding bandits considered in Appendix E. For the *E. coli* dataset, we obtain $\hat{\mathbf{v}}^{(r)}$ in Algorithm 1 using the scheme proposed in Baharav et al. [5], that is by taking column sums of $X^{(r)}$. We run our simulations on reference read 1 of their dataset, as reference read 0 has 10 non-trivial alignments, whereas reference read 1 has 5 as desired.

While in theory we do not reuse old samples to maintain independence, in our algorithm we do. This is done naturally by running Algorithm 1 on the $\{0,1\}^{|\mathcal{I}_r| \times \sum_{i=0}^{r} t_r}$ matrix of responses on all the previously asked questions (not just those in the current round). Similarly, we do not split our matrix in 2 for Algorithm 1; we estimate $\hat{\mathbf{v}}^{(r)}$ from the entirety of $X^{(r)}$, and compute $\hat{\mathbf{u}}^{(r)} = X^{(r)}\hat{\mathbf{v}}^{(r)}$.

For top 2k, we ran both algorithms to return their estimated top 2k, which we denote as the set Top-2k, then evaluated the performance as $\frac{1}{k}\sum_{i=1}^{k} \mathbb{1}\{(i) \in \text{Top-2k}\}$. For the error probability plot, we evaluated the performance by running the algorithms to return their estimated top k, Top-k, and computing $\mathbb{1}\{\text{Top-k} \equiv \{(i) : i \in [k]\}\}$. For each simulation, we run 100 trials and report the mean of our performance metric as well as its standard deviation (shaded in).

For the controlled experiments, $p_i$'s are generated according to a Beta$(1, 5)$ distribution and $q_i$ are generated according to a uniform$(0, 1)$ distribution.

## I.1 Computing architecture and runtime

Min-hashes took 18 hours to generate on 50 cores of an AMD Opteron Processor 6378 with 500GB memory. Generating the empirical results for uniform and adaptive on the *E. coli* dataset took took 36 minutes on one core. Generating the empirical results for the synthetic crowdsourcing experiments took 3 hours on one core, due to the fact that there is no efficient approximation for the right singular vector, and so one needs to compute the actual SVD of $X^{(r)}$ in every iteration.