[Reviews · NeurIPS 2020]

Review 1

Summary and Contributions: The authors discuss a spectral method for identifying the pairs of reads with the most likely significant overlaps between them, especially in the setting of long reads such as PacBio. This method is able to produce confidence intervals. Furthermore, the results on at least one biological dataset appear promising.

Strengths: 1) The authors derive a theoretically clean framework of rank-one approximations (I saw it at RECOMB so was already familiar with it, but I do consider it a key strength). 2) The authors prove rigorous bounds on their confidence intervals (albeit dependent on unspecified constants). 3) The authors demonstrate promising results on an E. coli genome sequenced with PacBio.

Weaknesses: 1) The paper feels unfinished and crammed; instead of including so much tecnhical detail, with derivations, in the main manuscript, I wish the authors had focused on the application, and summarized the technical details, relegating them to appendices. 2) The guarantees derived are unclear as they contain arbitrary constants, which are furthermore different between different theorems and lemmas. It is not clear what the results mean in practice, since asymptotics do not apply so readily to the situation of finite-length reads from a finite-length genome. 3) The conceptual jumps back-and-forth between workers, questions, reads, and k-mers are confusing; the paper needs to be greatly improved in terms of its clarity.

Correctness: As far as I was able to follow them (which is, not all the way), I believe that the claims are correct. The empirical methodology is promising, but comparisons are lacking.

Clarity: Not clear enough, there is a lot of room for improvement.

Relation to Prior Work: Instead of MHAP, I would have liked to see a more recent reference such as Mash.

Reproducibility: Yes

Additional Feedback:


Review 2

Summary and Contributions: The paper brings ideas from crowdsourcing and multi-armed bandit domains to the problem of pairwise alignment of biological sequences, viewing random hash functions as outputs of crowdsourcing workers, and solving it via spectral decomposition of a response matrix (k-mers taken from the biological sequences times number of hash functions). Where there are dominant alignments, a multi-armed bandit formulation helps refine this solution efficiently.

Strengths: The strength of the paper is in the crowdsourcing solution, the derivation of the spectral algorithm for identifying the top k items and theorem on the failure probabilities.

Weaknesses: But I am very disappointed with the paper from the point of view of what it promises in the title as algorithms for biological sequence analysis. Following a lot of derivations and theorems, the work in the biological domain is a simple tag-on in the last six lines of the paper with very little analysis of its empirical usefulness in practice. Since I am no expert in crowdsourcing algorithms and am reading this paper from the point of view of what it promises, it is disappointing and I do not vote for its acceptance in this conference. I hope the theoretical strand reviewers might support this on the basis of originality of contribution there.

Correctness: The algorithmic derivations appear correct.

Clarity: The paper is well written and easy to follow. But much of its content is not in line with what it promises.

Relation to Prior Work: Well described for the algorithmic part, but needs to address better the significance and relevance to biological sequence analysis.

Reproducibility: Yes

Additional Feedback: Discussion stage comment: I am the most negative about this submission, and happy to be ruled out as outlier. BUT, my reservation stands. A highly theoretical work being masqueraded as an application or potential application continues to disappoint me. I do not see an argument as to WHY such a method is suitable for sequence alignment. What assumption about biology / rules of evolution / characteristics of the measurement instrument gives us reason to believe that the method advanced is suitable? Or i it just an accident this is the way forward?? If it is the latter, the empirical work should be better done than a tiny paragraph hidden in volumes of theoretical work. BUT of course the theoretical part (which I have not gone through in detail) alone could make it a good paper -- in which case it should be pitched and evaluated as such.


Review 3

Summary and Contributions: In bioinformatics, is it of importance to identify similar regions between pairs of DNS sequencing reads. A current methodological trend consists of two steps, in that pairs those are likely to have large alignment are detected first, and expensive alignment algorithms are performed only on the candidates. For solving the first step problem, the authors employ two key ideas; a rank-one crowdsourcing model and a multi-armed bandit algorithm. The authors give theoretical analysis on the performance of the proposed algorithm as well as experimental results using both synthetic and real dataset. ---after rebuttal comments--- This reviewer has read the authors' response, and the comments from other reviewers. Considering all that, this reviewer would like to remain the original decision.

Strengths: The combination of two key ideas are novel.

Weaknesses: Computational experiments are rather limited. There is only one baseline method being compared, and the dependence of the proposed method on parameters are not examined much.

Correctness: Not sure. I have not checked the proofs.

Clarity: The paper is clearly written.

Relation to Prior Work: Related works are surveyed and described sufficently.

Reproducibility: Yes

Additional Feedback:


Review 4

Summary and Contributions: The authors consider minhash in a rank-1 crowd sourcing framework, solvable via spectral decomposition. A multi-armed bandit algorithm is then used to improve estimation via iterative refinement. A key application for the authors is pairwise alignment of DNA sequences with similarity estimated through the Jaccard similarity of k-mer sets.

Strengths: The presented bandit algorithm is well justified and demonstrated to have substantial computational advantages in the empirical section. The problem is broadly applicable, particularly in the biological domain. This is a good theoretical contribution with solid applications. Due to being broadly applicable and also computational efficient it is very relevant to the community.

Weaknesses: The authors present a thresholded bandit in the supplementary due to space limitations but no experiments involving it are presented in either the main text or supplementary. Would have been nice to see some results for this (even if only in the sup).

Correctness: The paper is solid. The proofs are easy to follow and the empirical methodology is good.

Clarity: The paper is well written and the results quite well presented. Figure 2 would be a bit more readable if the ribbon was 95% CIs rather than 1 sd.

Relation to Prior Work: The paper is well placed in existing literature. The novel contributions made by this article are clear.

Reproducibility: Yes

Additional Feedback: Will software be made available? I'd love to see this used in genomics algorithms, there are many applicable areas.

[Author Response · NeurIPS 2020]

We would like to thank all the reviewers for their thoughtful comments, especially given the difficult times. Your individual comments are addressed below (itemized by reviewer number and a short description of the comment).

**R2 (Empirical usefulness):** We will clarify parts of the introduction to make sure the paper does not promise more than it delivers. The goal of our paper is to establish a formal connection between min-hash based pairwise alignment and a rank-one crowdsourcing model and show how such a model can be efficiently solved using adaptivity. Our main motivating application is the pairwise alignment of third-generation (PacBio and ONT) sequencing data. Since these technologies produce noisy reads, most practical assembly pipelines employ min-hash based schemes to perform pairwise alignment [5,6,28] (citation numbers from the original manuscript). In particular, it has been previously shown by Baharav et al. [5] that spectral methods can be used to improve pairwise alignment accuracy via extensive experimentation on PacBio datasets from the NCTC 3000 project. Hence, our focus was on deriving a framework to compute confidence intervals for this spectral estimator, which allows bandit algorithms to be used to speed up the estimation. In the revised version, we will provide results on additional sequencing datasets and results based on the thresholding bandits, thus providing additional experimental validation of our approach. Moreover, following the comments of R4, we will be making our code available to maximize the potential practical impact.

**R1 (Paper Organization):** Following R1's suggestion, we will revise the paper for clarity and move some of the technical content from Section 3 into the appendix. This will free up space, which we will use for more experimental results. In particular, we will provide results on other PacBio datasets and results for the thresholding bandit algorithm.

**R1 (Unspecified constants):** While the constants were unspecified in our theorem statements, they can be computed explicitly based on the proofs in the appendix. In the revised paper, to maintain clarity, we will keep the constants unspecified in the main text, but will include detailed re-statements of the main theorems in the appendix with explicit constants. We will also clarify in the main text that these constants are nonasymptotic and point the reader to the appendix for their specific values. We will also include a remark explaining that the derived constants are quite loose, and that, in practice, a standard trick for improving performance is to first run the method on a dataset with ground truths to better approximate tighter (empirical) constants and then use these on the real problem.

**R1 (Consistency of terminology):** To avoid the back and forth between workers, questions, reads, and k-mers, we will discuss the connection with crowdsourcing in the Introduction, and then keep the rest of the discussion in terms of read alignments and hash functions.

**R1, R3 (Experiments):** Since the main purpose of the paper was to introduce adaptivity in the context of spectral estimation of pairwise alignments, the natural baseline method to compare with is the non-adaptive one. This emphasizes the gains obtained from adaptivity. We will include comparisons on two more datasets from the NCTC 3000 project (one of them, NCTC 4174, is shown in figures (b) and (c) below). These experiments focus on the gains provided by adaptivity, as the empirical usefulness of spectral methods for pairwise alignment had been previously studied in [5].

**R1 (Related Works):** We thank the reviewer for pointing us to Mash. We will add references and a short discussion regarding more recent alignment methods such as Mash and Mashmap in the revised version of the paper.

**R4 (Thresholding Bandits):** We thank the reviewer for the suggestion. We will present some of our discussion on threhsolding bandits in the main body of the paper, and we will include some thresholding bandits results in the Empirical Results section. In figure (c) above we show thresholding bandits results for the crowdsourcing problem (the set-up is identical to that of Fig. 2(c) in the paper but the task is to return a list that includes all products liked by at least 65% of the population and none liked by less than 50% of the population). As we see, the achievable points of the thresholding bandit algorithms are significantly better than the non-adaptive algorithm (the non-adaptive algorithm is the probability of error obtained at different fixed budgets and hence has no CIs around the number of workers).

**R4 (Figure 2):** We will change the ribbons to be 95% CI instead of 1 standard deviation in our plots.

**R4 (Software):** We thank the reviewer for the enthusiasm regarding software for our method. We plan on making our codebase publicly available once the review period is over.

[Meta-Review · NeurIPS 2020]

Three referees support accept, one indicates reject (with weak self-assigned confidence tough). In my opinion, the rebuttal nicely addressed all points of criticism raised, so I also recommend acceptance. However, please include the additional experiments from the rebuttal in your revised paper.